*Method*

# Measuring glycolytic flux in single yeast cells with an orthogonal synthetic biosensor

Francisca Monteiro[1,†,§], Georg Hubmann[1,‡,§], Vakil Takhaveev[1], Silke R Vedelaar[1], Justin Norder[1], Johan Hekelaar[1], Joana Saldida[1], Athanasios Litsios[1], Hein J Wijma[2], Alexander Schmidt[3] & Matthias Heinemann[1,*] iD

## Abstract

Metabolic heterogeneity between individual cells of a population harbors significant challenges for fundamental and applied research. Identifying metabolic heterogeneity and investigating its emergence require tools to zoom into metabolism of individual cells. While methods exist to measure metabolite levels in single cells, we lack capability to measure metabolic flux, i.e., the ultimate functional output of metabolic activity, on the single-cell level. Here, combining promoter engineering, computational protein design, biochemical methods, proteomics, and metabolomics, we developed a biosensor to measure glycolytic flux in single yeast cells. Therefore, drawing on the robust cell-intrinsic correlation between glycolytic flux and levels of fructose-1,6-bisphosphate (FBP), we transplanted the *B. subtilis* FBP-binding transcription factor CggR into yeast. With the developed biosensor, we robustly identified cell subpopulations with different FBP levels in mixed cultures, when subjected to flow cytometry and microscopy. Employing microfluidics, we were also able to assess the temporal FBP/glycolytic flux dynamics during the cell cycle. We anticipate that our biosensor will become a valuable tool to identify and study metabolic heterogeneity in cell populations.

**Keywords** biosensor; fructose-1,6-bisphosphate; glycolytic flux; single cell; yeast
**Subject Categories** Biotechnology & Synthetic Biology; Metabolism; Methods & Resources
**Mol Syst Biol. (2019) 15: e9071**

## Introduction

Increasing evidence suggests that individual cells in a population can be metabolically very different (Nikolic *et al*, 2013; van Heerden *et al*, 2014; Solopova *et al*, 2014; Kotte *et al*, 2015; Takhaveev & Heinemann, 2018). Metabolic heterogeneity has been found, for instance, not only in microbial cultures used for biotechnological processes (Xiao *et al*, 2016), but also in cells of human tumors (Strickaert *et al*, 2017). Because metabolic heterogeneity is connected with productivity and yield losses in biotechnological production processes (Xiao *et al*, 2016), and in cancer with limited therapeutic successes (Robertson-Tessi *et al*, 2015), it is key to identify metabolic subpopulations and to understand their emergence.

Toward assessing metabolic heterogeneity, several novel experimental tools have recently been developed to measure metabolite levels in single cells (Qiu *et al*, 2019), e.g., by exploiting the autofluorescence of specific metabolites (Papagiannakis *et al*, 2016), Förster resonance energy transfer (FRET) (Hou *et al*, 2011), or metabolite-binding transcription factors (Mahr & Frunzke, 2016). For instance, transcription factor (TF)-based biosensors now exist to detect amino acids (Mustafi *et al*, 2012), sugars (Raman *et al*, 2014), succinate and 1-butanol (Dietrich *et al*, 2013), triacetic acid lactone (Tang *et al*, 2013), and malonyl CoA (Xu *et al*, 2014), partly enabled by the transplantation of prokaryotic metabolite-responsive TFs to eukaryotes (Ikushima *et al*, 2015; Li *et al*, 2015; Skjoedt *et al*, 2016; Wang *et al*, 2016; Ikushima & Boeke, 2017).

While measurements of metabolite levels in single cells are already useful, knowledge of metabolic fluxes in individual cells would often be more informative, as metabolic fluxes represent the ultimate functional output of metabolism. Fluxes serve as predictor of productivity in the development of cell factories (Nielsen, 2003) or as indicator of disease (Zamboni *et al*, 2015). Here, particularly knowing the flux through glycolysis would be valuable, as this flux has been shown to correlate with highly productive phenotypes (Gupta *et al*, 2017) and cancer (Pavlova & Thompson, 2016). While nowadays metabolic fluxes can be resolved in ensembles of cells, for instance, by means of $^{13}C$ flux analysis (Antoniewicz, 2015), inference of fluxes in individual cells, however, is not possible until today (Takhaveev & Heinemann, 2018).

1 Molecular Systems Biology, Groningen Biomolecular Sciences and Biotechnology Institute, University of Groningen, Groningen, The Netherlands
2 Biotechnology, Groningen Biomolecular Sciences and Biotechnology Institute, University of Groningen, Groningen, The Netherlands
3 Biozentrum, University of Basel, Basel, Switzerland
*Corresponding author. Tel: +31 50 363 8146; E-mail: m.heinemann@rug.nl; Twitter: @HeinemannLab
†Present address: cE3c-Centre for Ecology, Evolution and Environmental Changes, Faculdade de Ciências, Universidade de Lisboa, Lisboa, Portugal
‡Present address: Laboratory of Molecular Cell Biology, Department of Biology, Institute of Botany and Microbiology, KU Leuven, & Center for Microbiology, VIB, Heverlee, Flanders, Belgium
§These authors contributed equally to this work as first, second authors

One possible avenue toward measuring metabolic fluxes in individual cells has recently emerged by the discovery of so-called flux-signaling metabolites (Litsios *et al*, 2018), which are metabolites, whose levels—by means of particular regulation mechanisms (Kochanowski *et al*, 2013)—strictly correlate with the flux through the respective metabolic pathway. Such flux signals are used by cells to perform flux-dependent regulation (Kotte *et al*, 2010; Huberts *et al*, 2012). Biosensors for such metabolites, such as recently accomplished for *E. coli* (Lehning *et al*, 2017), would in principle allow for measurement of metabolic fluxes in single cells, in combination with microscopy or flow cytometry.

Here, drawing on the glycolytic flux-signaling metabolite fructose-1,6-bisphosphate (FBP) in yeast (Huberts *et al*, 2012; Hackett *et al*, 2016; preprint: Kamrad *et al*, 2019) and using the *B. subtilis* FBP-binding transcription factor CggR (Doan & Aymerich, 2003), we developed a biosensor that allows for sensing FBP levels, and thus glycolytic flux, in single yeast cells. To this end, we used computational protein design, biochemical, proteome, and metabolome analyses (i) to develop a synthetic yeast promoter regulated by the bacterial transcription factor CggR, (ii) to engineer the transcription factor's FBP-binding site toward increasing the sensor's dynamic range, and (iii) to establish growth-independent CggR expression levels. We demonstrate the applicability of the biosensor for flow cytometry and time-lapse fluorescence microscopy. We envision that the biosensor will open new avenues for both fundamental and applied metabolic research, not only for monitoring glycolytic flux in living cells, but also for engineering regulatory circuits with glycolytic flux as input variable.

## Results

### Design of biosensor concept

For our biosensor, we exploited the fact that the level of the glycolytic intermediate fructose-1,6-biphosphate (FBP) in yeast strongly correlates with the glycolytic flux (Christen & Sauer, 2011; Huberts *et al*, 2012). Furthermore, we used the transcription factor CggR from *B. subtilis*, to which FBP binds (Doan & Aymerich, 2003). When bound to its target DNA, CggR forms a tetrameric assembly of two dimers, through which transcription gets inhibited (Zorrilla *et al*, 2007b). Upon binding of FBP to the CggR–DNA complex, the dimer–dimer contacts of CggR are disrupted (Zorrilla *et al*, 2007a), which decreases the overall CggR/operator complex stability, leading to increased CggR dissociation, and thus derepression of the promoter (Chaix *et al*, 2010).

Here, we aimed to transplant the *B. subtilis* CggR to yeast and have it exerting FBP-dependent and thus glycolytic flux-dependent regulation of expression of a fluorescent protein. To this end, a number of challenges had to be addressed. First, a synthetic promoter had to be designed for the foreign transcription factor CggR, involving the identification of ideal positioning and number of operator sequences (Teo & Chang, 2014, 2015), and engineering the nucleosome architecture to allow for maximal promoter activity (Curran *et al*, 2014). Second, CggR had to be made responsive to FBP in the correct dynamic range, requiring protein engineering efforts (Raman *et al*, 2014; Rogers *et al*, 2015). Third, the CggR expression levels needed to be such that together with the metabolite-modulating effect on CggR, the TF can actually exert a regulating effect on the promoter, for which we needed to identify proper CggR expression levels (Fig 1).

### *In vivo* test system for a substrate-independent and growth rate-independent flux sensor

For later evaluation of the flux-reporting capacity of the developed sensor, we first established an *in vivo* test system, through which we could generate a range of glycolytic fluxes at steady-state conditions. To this end, we employed a combination of growth substrates and two different *S. cerevisiae* strains: the wild type (WT) and a mutant strain (TM6), which only carries a single chimeric hexose transporter and thereby only generates low

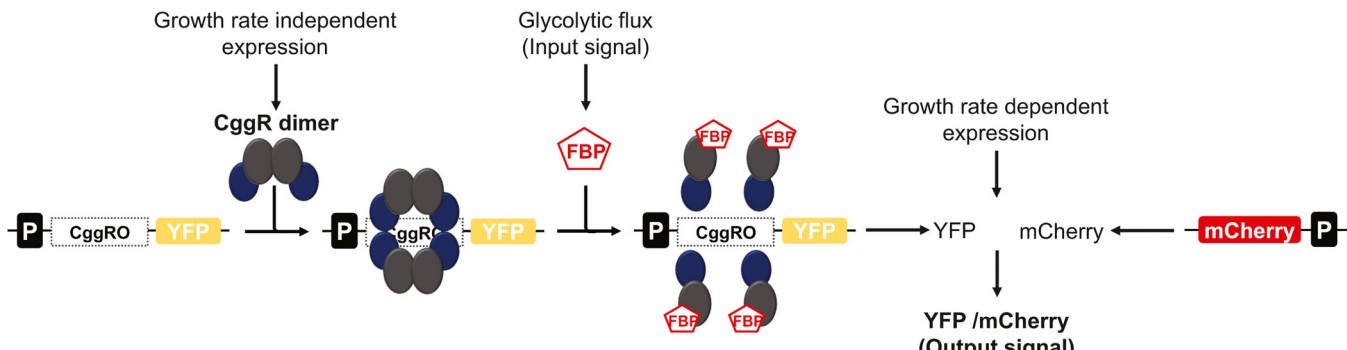

**Figure 1.  Illustration of the biosensor concept to measure glycolytic fluxes in single *S. cerevisiae* cells.**

Expression of the bacterial transcriptional repressor CggR at constant levels, i.e., independent of growth rate and substrates. Binding of CggR as a dimer of dimers to the operator (CggRO) of the synthetic cis-regulatory region, forming the CggR–DNA complex repressing transcription. At high glycolytic fluxes, fructose-1,6-bisphosphate (FBP) levels are high and FBP binds to CggR disrupting the dimer–dimer contacts, which induces a conformational change in the repressor, such that transcription of the reporter gene (YFP) can occur. The binding of FBP to CggR and consequent transcription is dependent on the FBP concentration, which correlates with glycolytic flux. The activity of the glycolytic flux biosensor is measured by quantifying YFP expression. YFP expression levels are normalized through a second reporter, mCherry, under the control of TEF1 mutant 8 promoter ($P_{TEFmut8}$), to control for global variation in protein expression activity.

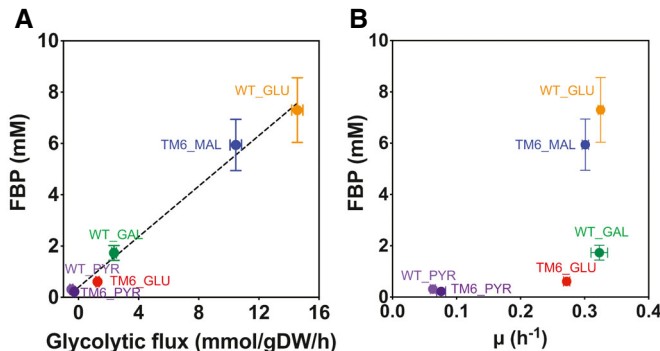

**Figure 2.  FBP concentration linearly correlates with glycolytic flux, stronger than with growth rate.**

A  Glycolytic flux of wild type (WT) and TM6 strains strongly correlates with fructose-1,6-bisphosphate (FBP) concentration. The glycolytic flux is reported here as the flux between the metabolites fructose 6-phosphate (F6P) and FBP. Glycolytic fluxes were obtained on the basis of physiological and metabolome data, and via a novel method to estimate intracellular fluxes (Niebel *et al*, 2019). While on high glucose, the WT strain accomplishes a high glucose uptake rate (and thus glycolytic flux), the mutant strain (TM6) only generates a low glucose uptake (and thus glycolytic flux). On maltose, also the mutant strain achieves a high glycolytic flux, since maltose is transported by a separate transporter (Chang *et al*, 1989).

B  FBP concentration as a function of cellular growth rate shows weaker correlation.

Data information: For metabolite levels and growth rates, error bars correspond to the standard deviation between three independent experiments, for glycolytic fluxes to the mean and standard deviations of the sampled flux solution space (cf. Materials and Methods). The carbon sources were used at a final concentration of 10 g/l and are indicated: glucose (GLU); galactose (GAL); maltose (MAL); and pyruvate (PYR). To assess the linear correlation between the FBP concentration and the glycolytic flux (A) or growth rate (B) across the studied conditions, we implemented Pearson's correlation analysis assisted by bootstrapping. Specifically, we used in total 53 FBP concentration measurements corresponding to six different metabolic conditions (combinations of strains and carbon sources), biological and technical replicates. We paired each of these FBP measurements with the mean and standard deviation of the model-derived glycolytic flux (A) or of the growth rate (B) in the corresponding metabolic condition. We assumed the normal distribution of the flux and growth rate with the given mean and standard deviation in every condition, and implemented ordinary non-parametric bootstrapping with 100,000 iterations by randomly sampling values with replacement from the 53 FBP measurements and flux or growth rate distributions to calculate the correlation statistics. In (A), Pearson's coefficient was found to be 0.97 with [0.95, 0.99] as the 95% confidence interval, and a *P*-value smaller than 2.23e-308 (normal bootstrap). In (B), Pearson's coefficient was found to be 0.73 with [0.64, 0.80] as the 95% confidence interval, and *P*-value equal to 2.28e-77 (normal bootstrap).

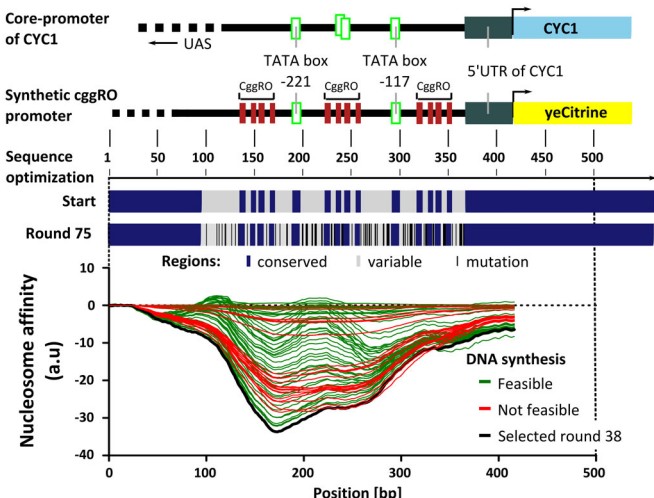

**Figure 3.  Design of the synthetic CggR cis-regulatory element.**

The promoter design is based on the *CYC1* core promoter. The relevant structural elements of the *CYC1* core promoter elements, which are required for transcription, were conserved in the synthetic promoter design. These elements comprised two TATA boxes at positions −221 and −117 (relative to the start of the *CYC1* ORF), and the 5′UTR of the *CYC1* core promoter (including transcriptional start site, TSS). In the promoter design, three CggR operator sites were inserted adjacent to the two TATA boxes. All functional elements were conserved (blue colored region) during the optimization of the promoter sequence. Nucleotide sequences between the functional elements (gray colored region) were allowed to be optimized by the algorithm. Nucleotides that got optimized are indicated with a black line. A total of 75 sequence versions were generated, where each sequence differed in one mutation from the progenitor sequence. The sequences were optimized for low nucleosome affinity. After optimization, all sequences were checked for synthesis feasibility. The synthesis of the sequences was feasible (green) until the 46[th] round. After this round, the sequences (not feasible in red) reached a GC content insufficient for proper synthesis. The promoter sequence, which was generated in round 38 (black), showed the best compromise between minimal nucleosome affinity and the possibility to synthesize the sequence.

## Development of the synthetic CggR cis-regulatory element

First, we designed a synthetic CggR cis-regulatory element for yeast (CggRO) based on the *CYC1* promoter, which was previously successfully re-designed (Curran *et al*, 2014). To accomplish repression of the promoter by CggR, we aimed to shield the TATA boxes by the binding and tetramerization of the CggR dimers. The *CYC1* core promoter has three TATA boxes at the positions −221, −169, and −117, upstream of the open reading frame (Fig 3—upper part). We flanked the two TATA boxes at positions −221 and −117 up- and downstream with a CggR operator site. To conserve the geometry of the *CYC1* core promoter as much as possible, we removed the TATA box at position −169, because this TATA box was exactly located where we integrated the CggR operator sites flanking the other TATA boxes, and we did not want to make the sequence longer. The 5′UTR of the *CYC1* promoter, which also included the transcriptional start site, was kept. To allow for sole binding and regulation through CggR, we removed the part further upstream of the TATA box at the position −221 where, according to YEAS-TRACT (Teixeira *et al*, 2014), the endogenous transcriptional binding sites of the *CYC1* promoter are located.

glucose uptake rates at high glucose levels (Elbing *et al*, 2004). Metabolome and physiological analyses in combination with a new method for intracellular flux determination (Niebel *et al*, 2019) showed that this combination of strains and conditions allowed us to generate a broad range of glycolytic fluxes (Fig 2A). Consistent with the earlier reported correlation between FBP levels and glycolytic flux (Huberts *et al*, 2012), also here the FBP levels had a strong linear correlation with the flux [*r* = 0.97, (0.95, 0.99) 95% confidence interval] (Fig 2A), but not with growth rate (Fig 2B). This set of conditions and strains served as test system for the to-be-developed glycolytic flux sensor.

Using a computational method (Curran *et al*, 2014), we further optimized this designed sequence of the CggR cis-regulatory element to minimize nucleosome binding. Functional elements (e.g., the CggR operator sites, the TATA boxes, and the 5′UTR; cf. Appendix Tables S1 and S2) were excluded from the sequence optimization (Fig 3—lower part). A total of 75 computational optimization rounds were applied. As the CggR cis-regulatory element resembled a repetitive DNA sequence with a high AT content, sequence variants were checked for DNA synthesis feasibility. The cis-regulatory element of round 38 was the variant with the lowest nucleosome affinity but with retained feasibility for DNA synthesis. The synthesized synthetic promoter was integrated upstream of the fluorescent reporter protein YFP (eCitrine) in a centromeric plasmid ensuring a stable copy number.

## Establishing a substrate-independent and growth rate-independent CggR expression

Next, to drive expression of CggR, we needed a promoter that would lead to condition-independent (i.e., constant) intracellular CggR levels in order to ensure that the flux sensor only reports altered FBP levels (i.e., glycolytic fluxes), and not altered CggR levels. To this end, we tested the $P_{CMV}$ promoter, which is widely used as a strong constitutive promoter in mammalian cells (Boshart *et al*, 1985), and two mutant variants of the endogenous TEF1 promoter, i.e., mutant 2 ($P_{TEFmut2}$) with low, and mutant 7 ($P_{TEFmut7}$) with medium-to-high expression strength (Nevoigt *et al*, 2006). Each promoter and the CggR gene were cloned into the HO genomic locus of both yeast strains.

To quantify the CggR protein levels, we performed proteome analyses with the different strains, promoters, and growth conditions. Overall, the three promoters yielded largely different CggR abundances on glucose (Fig 4A). Across conditions and growth rates, we found that the CggR levels when expressed from the $P_{CMV}$ and $P_{TEFmut2}$ promoters showed significant variations, while the $P_{TEFmut7}$ promoter generated more comparable CggR levels across growth rates (Fig 4B), as established through the different carbon sources and strains. Because of its more condition-independent expression level, we selected the $P_{TEFmut7}$ promoter to drive the CggR expression.

## Engineering the FBP affinity of CggR

Next, we needed to engineer the FBP binding to CggR, such that it matches with the physiological range of FBP levels. FBP levels in yeast range from 0.2 mM to around 8 mM (Fig 2A). As the wild-type CggR has an affinity for FBP of around 1 mM (Bley Folly *et al*, 2018), we needed to generate a CggR mutant with a slightly lower affinity for FBP, and with ideally a graded interaction between CggR and FBP toward accomplishing a broad dynamic response range of the sensor. Importantly, the engineered CggR would still need to bind to the DNA, and furthermore, the protein should be stable to not affect its cellular abundance.

To obtain such a CggR mutant, supported by computational protein design methods, we identified mutations at the CggR–FBP-binding site that could lead to the desired decrease in affinity. Specifically, as in the CggR structure (3BXF) (Rezácová *et al*, 2008) CggR binds to FBP through hydrogen bonds, and we

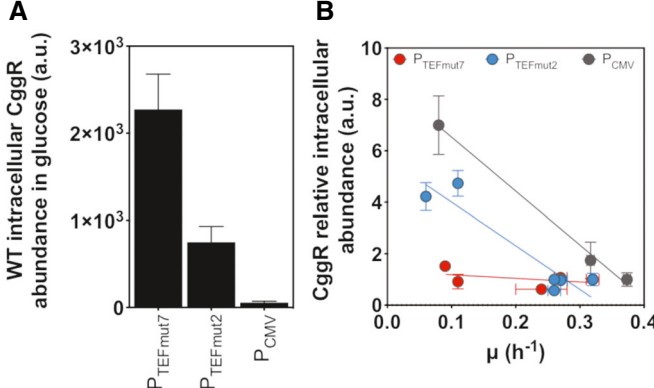

**Figure 4. CggR intracellular levels and expression profile with different promoters, strains, and conditions.**

A   CggR intracellular abundance in the wild-type (WT) strain on glucose strongly varies with the promoter used. The CggR intracellular levels were quantified by proteomics in steady-state cultures grown in minimal media with glucose as carbon source at a final concentration of 10 g/l. Error bars represent the standard deviation of at least three replicate experiments.

B   The relative abundance of CggR (normalized to the abundance measured on glucose and the same promoter) is almost constant with $P_{TEFmut7}$ across multiple growth rates in WT and TM6 cells, but not with $P_{TEFmut2}$ and $P_{CMV}$. The CggR intracellular levels were quantified by proteomics in steady-state cultures grown in minimal media with glucose, galactose, maltose, or pyruvate as carbon sources at a final concentration of 10 g/l. WT data include all three promoters, whereas TM6 only includes the $P_{TEFmut2}$ and $P_{TEFmut7}$ data. Error bars represent the standard deviation of at least three replicate experiments.

designed mutations to weaken or disrupt H-bonding interactions (Table 1, Appendix Table S3), with the aim to decrease binding affinity. The X-ray structure further showed that FBP binding causes a conformational change in CggR (Rezácová *et al*, 2008), where a loop between residues G177 and Q185 moves away from the FBP-binding site toward another subunit. On the basis of this, we conjectured that mutations might not only influence FBP binding, but also alter the equilibrium between the normal and activated conformation, even in the absence of FBP. To predict the effect of the mutations on this equilibrium, and on overall protein stability, we used FoldX (Guerois *et al*, 2002), where we found that a E269Q mutation could decrease overall stability while R175K could permanently shift CggR to its activated conformation (Table 1). Four mutations (i.e., T151S, T151V, T152S, and R250A) were thus identified as promising candidates for decreasing the FBP binding to CggR without otherwise negative effects (Table 1).

We generated these CggR mutants with site-directed mutagenesis, expressed in *E. coli*, purified, and biochemically characterized them. To this end, we used thermal shift assays to assess protein stability and ligand binding. Most of the engineered CggR variants maintained wild-type stability, with the exception of E269Q (consistent with the above analysis) and T151V, which were less stable as indicated by decreased melting temperatures (Fig 5A). While the wild type had a $K_D$ of 1 mM FBP, the mutants T151S, E269Q, and T152S showed a 1.1-, 1.5-, and 1.6-fold lower $K_D$ values, respectively, while the $K_D$ values of the R250A and T151V mutants increased 1.5- and 2.6-fold (Fig 5B).

**Table 1. List of predicted mutations to alter CggR-FBP-binding affinity and stability.**

| Mutation | Expected effects on affinity for FBP (and if relevant on equilibrium and stability)[a] | FoldX predicted stability changes (kJ/mol)[b] | | |
| --- | --- | --- | --- | --- |
| | | $\Delta\Delta G^{fold}$ for the normal conformation | $\Delta\Delta G^{fold}$ for the activated conformation | $\Delta\Delta\Delta G^{fold}$ (between the conformations) |
| T151S | Negligible to mild affinity decrease | −1.4 | −1.3 | 0.1 |
| T151V | Mild to strong affinity decrease | 2.5 | 3.1 | 0.5 |
| T152S | Negligible to mild affinity decrease | 7.3 | 4.6 | −2.7 |
| R175K | Negligible to strong affinity decrease and possibly a shift of equilibrium to the activated conformation | 16.5[c] | −0.7 | −17.2[c] |
| R250A | Mild to strong affinity decrease | 2.2 | 2.7 | 0.5 |
| E269Q | Mild to strong affinity decrease for FBP in combination with overall destabilization | 16.4[c] | 11.4[c] | −5.0 |

[a]A detailed justification for the expected effects of the mutations on binding affinity is given in Appendix Table S4.
[b]A downshift in $\Delta\Delta G^{fold}$ predicts stabilization of the protein, while a downshift in $\Delta\Delta\Delta G^{fold}$ predicts that the FBP conformation becomes more favorable. $\Delta\Delta\Delta G^{fold}$ represents the difference between the $\Delta\Delta G^{fold}$ values for the two conformations.
[c]Values are significantly higher than the standard deviation of FoldX predictions, which equals 3.4 kJ/mol (Guerois *et al*, 2002).

To assess the DNA-binding capacity of the generated mutants, we performed electro-mobility shift assays. We first measured the percentage of CggR bound to DNA in the absence of FBP, reflecting CggR's binding affinity to DNA. Here, we found that the mutants R175K and E269Q variants did not bind to the DNA anymore (Fig 5C), and the mutant T151V only bound with lower affinity. The other mutants had a comparable DNA binding as the wild-type CggR. Next, using high (i.e., saturating) FBP levels (20 mM) to maximally promote release of CggR from the DNA and comparing the ratio between the percentage of the CggR bound to DNA, obtained at 0 mM of FBP, divided by the bound fraction at 20 mM, we found that only the R250A variant behaved similarly to the wild type, with around 30% of the CggR remained bound to DNA at high FBP levels (Appendix Fig S1). A comparison of the predicted mutant features with the actually observed ones is shown in Appendix Table S4.

Thus, as the R250A mutant fulfilled all desired criteria (Fig 5D), i.e., it showed the desired decrease in FBP affinity, had a similar stability and DNA-binding capability as the wild-type CggR, we selected this mutant for the sensor. This mutant had the additional advantage that it showed a flattened sigmoidal binding curve (Fig 5A, R250A), which is ideal for a sensor that needs to respond to a broad (cf. Fig 2A) FBP concentration range. The R250A mutation eliminated the arginine side chain that made two H bonds with the 6-phosphate group of FBP in the wild-type structure (Fig 5E) and replaced it with a hydrophobic alanine side chain, which cannot make H bonds.

**Testing the glycolytic flux sensor**

We implemented the flux-sensor elements in the wild type and mutant (TM6) strains using either the wild-type CggR or the CggR R250A mutant, genomically integrated into the HO locus under the control of the TEF1 promoter mutant 7 ($P_{TEFmut7}$). We added a centromeric plasmid with the cis-regulatory CggR element (CggRO) controlling YFP (eCitrine) expression. To normalize the YFP signals for extrinsic cell-to-cell variation in the global state of the protein expression machinery, we added the second fluorescent reporter protein RFP (mCherry) to the plasmid, under the control of the constitutive $P_{TEFmut8}$ promoter (Nevoigt *et al*, 2006). Through proteome analyses, we confirmed that CggR and mCherry expression levels correlate (Appendix Fig S2), validating the use of mCherry to normalize YFP expression. To determine the sensor output, i.e., the CggR activity, we quantified the YFP and mCherry fluorescence levels by flow cytometry. We did not perform any spectral compensation as spectral overlap is basically absent with the applied fluorophores and filters (Appendix Fig S3). The ratio between the YFP and mCherry fluorescence, each corrected for autofluorescence determined by FACS, yielded the CggR repressor activity.

First, we investigated which of the steps in the promotor engineering were influencing the activity of the CggR cis-regulatory element. To this end, we tested four promoter variants (Fig 6A, Appendix Fig S4): (i) the wild-type *CYC1* core promoter before the introduction of the CggR cis-regulatory elements, (ii) the *CYC1* promoter with the introduced CggRO elements, (iii) the *CYC1* promotor with the introduced CggRO elements after the initialization of the optimization algorithm for the nucleosome positioning (v1), and (iv) the *CYC1* promoter with the introduced CggRO elements, after optimization of the nucleosome affinity (v38), which differs in 37 positions from (iii) (Appendix Fig S4). Here, we found that the variants (i) to (iii) showed YFP fluorescence that is hardly above the background fluorescence, while the promoter with the optimized (i.e., lowest) nucleosome affinity showed much higher expression levels (Fig 6B). The ratios between the YFP and mCherry fluorescence for the different promoter variants underline the much increased activity of the promoter with optimized nucleosome affinity (Fig 6C). These results show that the sequence optimization was indeed necessary.

Next, we used the engineered and nucleosome-optimized variant of the promoter for both the wild type and mutant (TM6) strains and grew these strains on the different carbon sources. Growth rate analyses demonstrated that expression of the sensor constructs did not alter growth (Appendix Fig S5) and titration effects can also be

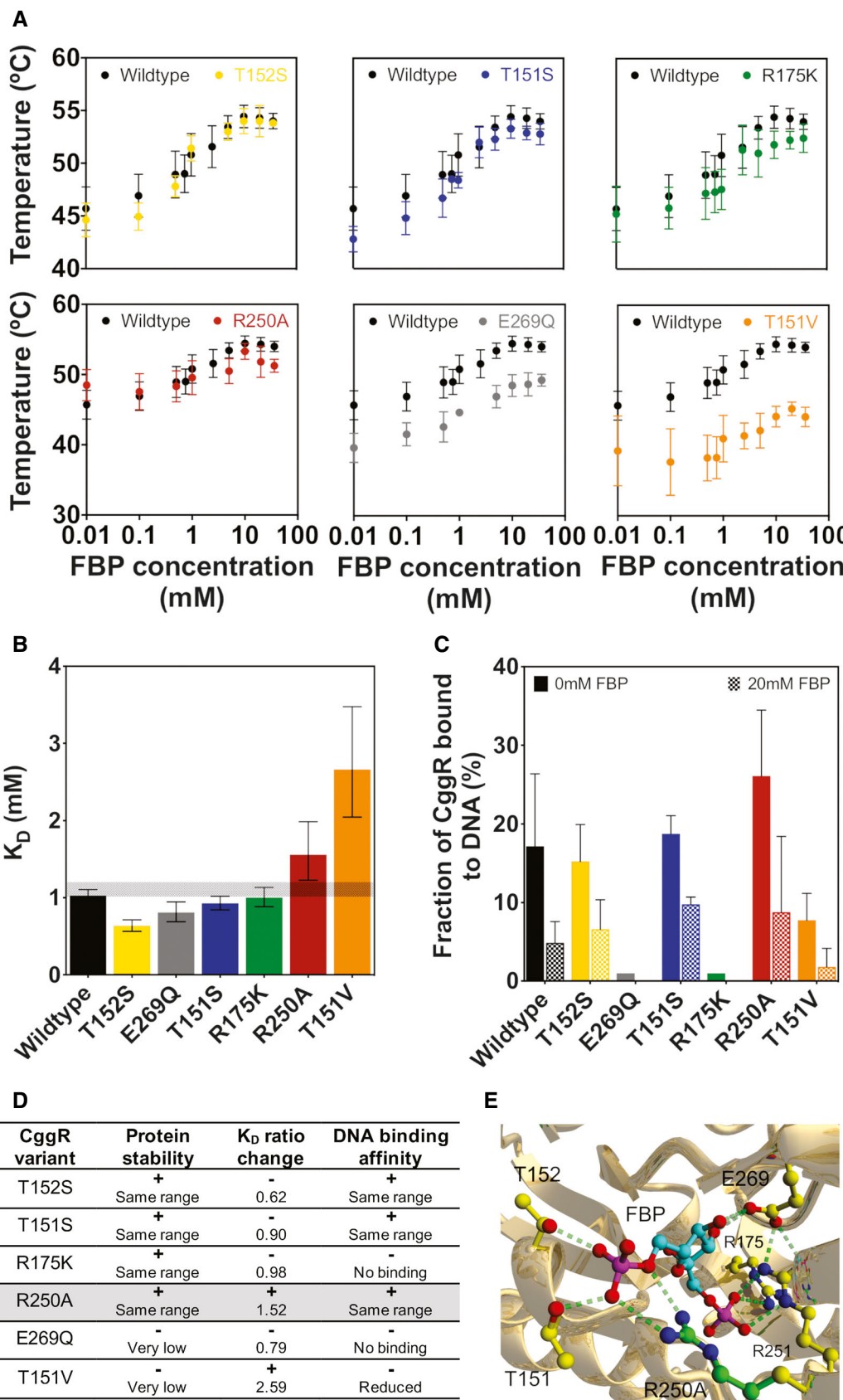

**Figure 5.**

**Figure 5.  Biochemical characterization of CggR and respective mutants.**

A  Thermal shift assays were used to determine the melting curves of wild-type CggR and mutants. Error bars correspond to the standard deviation of at least five replicates.

B  CggR-FBP affinity constants ($K_D$'s) of the wild type and mutant variants, determined by fitting a simple cooperative binding model to the melting curves data. Error bars indicate the 95% confidence intervals.

C  Quantification of CggR binding to DNA. The CggR bound to DNA fraction was calculated by dividing the intensity of the protein–DNA complex band by the total DNA. The background-subtracted total intensities of the CggR–DNA complex and the free-DNA bands were assessed with ImageJ. Error bars correspond to the standard deviation of three replicate experiments for the mutants and six for the wild type.

D  Summary of the biochemical characterization of the wild type and mutant CggR variants. $K_D$ ratio change indicates the ratio between the $K_D$ of the CggR mutant variant and one of the wild type. A plus sign indicates desired characteristics achieved in the mutants; a minus sign indicates undesired effects of the mutations. The mutant highlighted with gray background is the one we selected for further analyses, as this mutant had the desired characteristics with regards to all three criteria.

E  Ray-traced picture of the wild-type CggR and R250A mutant structure. Carbon atoms of the FBP ligand are in turquoise while the part of the side chain that is eliminated by the R250A mutation is in green. Hydrogen bonds are indicated with dashed lines.

excluded as CggR copy numbers are orders of magnitude lower than the cellular FBP copy numbers. Here, consistent with our design concept and the expected FBP-dependent derepression of the synthetic promoter, we found a strong positive correlation between the YFP/mCherry ratios and the intracellular FBP levels (Fig 6D) and glycolytic flux (Fig 6E). This correlation was absent in control strains lacking CggR (Fig 6D and E). Furthermore, consistent with the growth rate-independent design of the sensor to respond solely to FBP levels, we found no correlation between the sensor output and the growth rate (Appendix Fig S6).

While the sensor with the wild-type CggR (i.e., not optimized for FBP-binding affinity) displayed also a correlation with FBP levels, the optimized version (in line with its lower FBP affinity) displayed a dynamic response that better covered the physiological concentration range of FBP (Fig 6D) and thus has a better capability to distinguish different glycolytic flux values. When we estimated the wild type and R250A CggR fraction bound to FBP, we observed that the main differences occurred at intermediate FBP levels (between 1.5 and 2.5 mM), in agreement with the fact that the $K_D$ values of the two CggRs are around these FBP concentrations (Fig 6F).

Notably, the single point mutation in CggR (R250A) led to a different response curve (cf. Fig 6D), which is consistent with the fact that the FBP affinity of CggR is in the range of the physiological FBP concentrations, where small changes in the $K_D$ value lead to significant changes in the response. Further, as the point mutation solely altered the affinity to FBP (Fig 5B), but not its DNA affinity (Fig 5C), stability (Fig 5D) nor its cellular abundance (Appendix Fig S7), this demonstrates that the sensor's output exclusively depends on the changing FBP levels. Thus, these data demonstrate that we have generated a sensor for FBP and, as FBP levels correlate with glycolytic flux (Fig 2A), a sensor that robustly and specifically reports glycolytic flux. While the wild type and the mutant (TM6) strains used here have very different glycolytic flux levels during growth on glucose (Fig 2A), notably, our sensor unmasks this difference even though the environment was identical.

Altogether, this demonstrates that the recorded fluorescence ratio specifically responds to FBP levels. Because of its correlation with glycolytic flux (Huberts *et al*, 2012), this means that we have generated a sensor that reports glycolytic flux. In cases where the glycolytic fluxes are expected to change over a broad range, the use of the mutant CggR is most advisable, but in cases where high resolution is needed at low glycolytic fluxes, the use of the wild-type CggR might be preferred.

## Application of the sensor

Toward testing and applying the sensor, we first asked whether we could detect subpopulations with different glycolytic fluxes with flow cytometry. To this end, we mixed wild type and TM6 cells grown on glucose at different proportions. By plotting the single-cell signals from the YFP against the mCherry channel, we could clearly identify two clouds corresponding to the two strains (Fig 7A). Histograms over the single-cell YFP/mCherry ratios showed that with the glycolytic flux difference as present between the wild type and the TM6 cells on glucose, subpopulations with a minimal fraction of about 5% can be discovered with flow cytometry (Fig 7B).

Next, we aimed to test the engineered flux sensor with regard to its capability to detect single-cell differences in glycolytic flux when using microscopy, offering the possibility to co-assess other parameters, such as growth and cell division. First, we confirmed that also with the microscopic setup the output of the flux sensor still displays a linear correlation with glycolytic flux across conditions and strains (Fig 7C, Appendix Fig S8A). Next, we used microfluidics and time-lapse microscopy to cultivate the TM6 strain on glucose, where we recently showed that dividing cells with high flux co-exist with a small isogenic fraction of non-dividing cells with low glycolytic flux levels (Litsios *et al*, 2019). Here, using the sensor, we found that non-dividing cells had indeed significantly lower YFP/mCherry ratios, even visibly by eye, compared with their co-existing dividing counterparts (Fig 7D and E, Appendix Fig S8B), in line with their lower glycolytic flux. These results demonstrate that our flux sensor can be also used with microscopy, and is thus suitable for discrimination of individual cells with regard to their glycolytic flux levels, even within clonal cell populations.

To investigate whether the engineered biosensor can also be applied to study metabolic dynamics in single cells, we employed it to assess the FBP concentration, and thus the glycolytic flux, during the cell cycle. We cultivated TM6 cells with the sensor on high glucose in the microfluidic device, continuously measured with microscopy YFP and mCherry signals as well as cell volume, and identified budding and cytokinesis to demark cell cycles and their phases. Toward obtaining a biosensor readout suitable for reporting momentary FBP levels during the cell cycle, we abandoned the ratio of the YFP and mCherry signals since these signals result from the fluorescent-protein expression over a long period of time. Instead, we determined the momentary YFP and mCherry production rates, and used their uncoupling as a proxy for momentary FBP concentration.

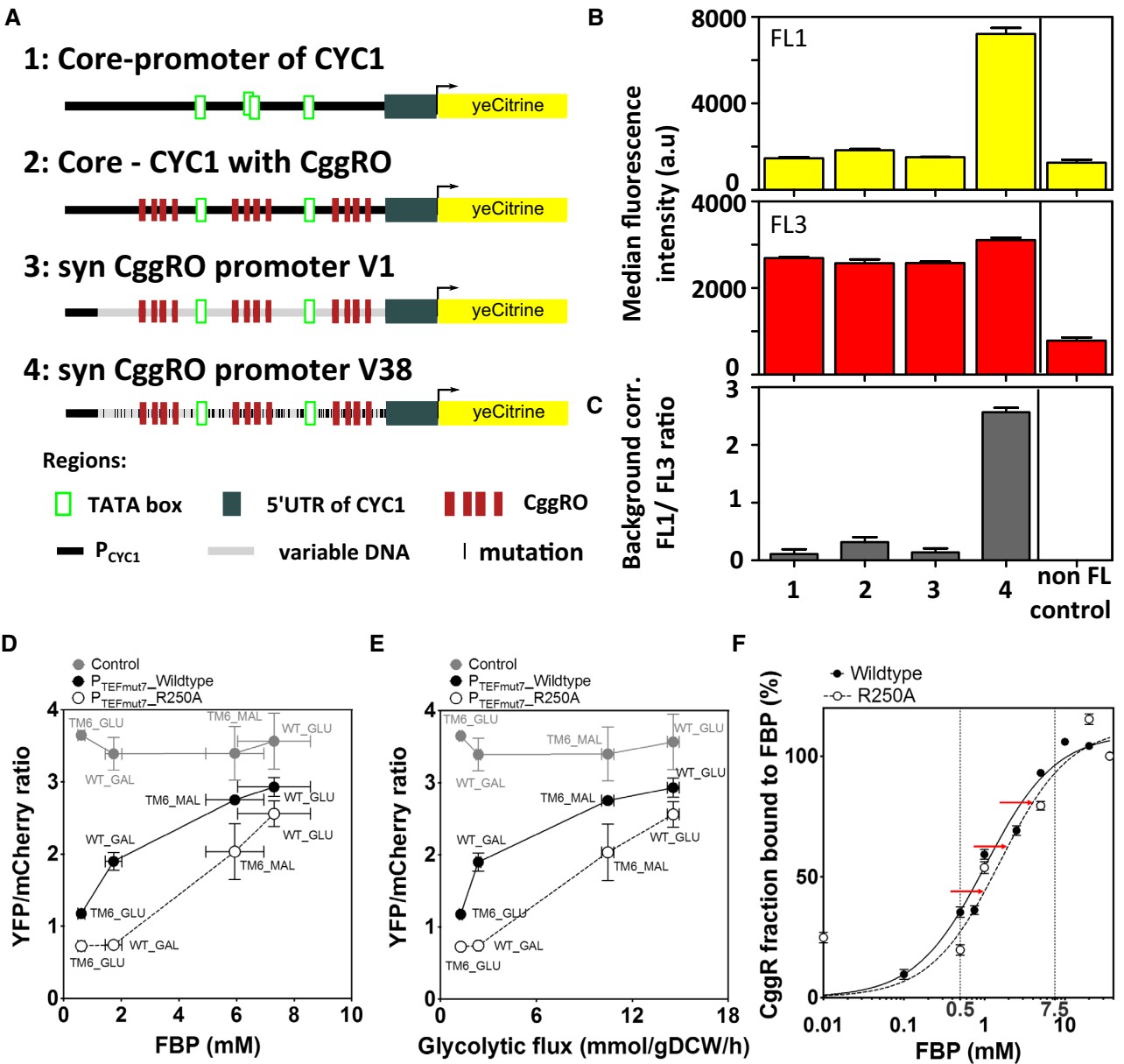

**Figure 6. The engineered flux-sensor reports glycolytic flux with a high dynamic flux range.**

A   Overview about the different design steps in our promotor engineering strategy (cf. also Appendix Fig S4).

B   The four reporter plasmids were transferred to the wild-type strain containing the CggR (R250A) under the control of the $P_{TEFmut7}$. The strength of the four promoters was assessed by quantifying YFP (FL1) and mCherry (FL3) fluorescence in exponentially growing wild-type cells in minimal medium with 10 g/l glucose. The FL1 and FL3 fluorescence shown is the non-background-corrected median of 100,000 cell events. The non-FL control is the signal from a wild-type strain grown under the same conditions. Error bars represent the standard deviation of three independent experiments.

C   The background fluorescence, assessed by the wild-type harboring the YCplac33 plasmid, was subtracted from FL1 and FL3. The final reporter activity is the ratio of the background-corrected YFP and mCherry values. Error bars represent the standard deviation of three independently determined ratios from three replicate experiments.

D, E   Reporter activity of the sensor across (D) multiple FBP levels and (E) glycolytic fluxes. The glycolytic flux is reported as the flux between the metabolites fructose 6-phosphate (F6P) and fructose-1,6-bisphosphate (FBP). Glycolytic fluxes were here estimated on the basis of physiological and metabolome data and a novel method to estimate intracellular fluxes (Niebel *et al*, 2019). Reporter activity is given by the YFP/mCherry ratio, calculated through the quantification of YFP and mCherry fluorescence along culture time using flow cytometry. Both YFP and mCherry fluorescence levels were first corrected for background using the same strains harboring the YCplac33 plasmid (Appendix Table S8). The control is the wild type and TM6 strains expressing only the reporter plasmid without CggR. Error bars represent the standard deviation of at least three replicate experiments.

F   Fraction of CggR bound to FBP across FBP concentrations. The red arrows indicate the shift in the percentage of CggR bound to FBP achieved in the R250A variant. The percentage of CggR molecules bound to FBP was calculated after normalizing the $T_m$ values for unbound/bound state using the $T_m$ at 0 mM FBP as unbound and at 36 mM (corresponding to maximum FBP concentration used) as total bound states. The curve fitting of the normalized values of CggR fraction bound to FBP was performed using a one-site specific binding model in GraphPad. The solid line corresponds to the wild-type CggR and the dashed line to the R250A variant. Vertical lines delimit the physiological FBP range.

   

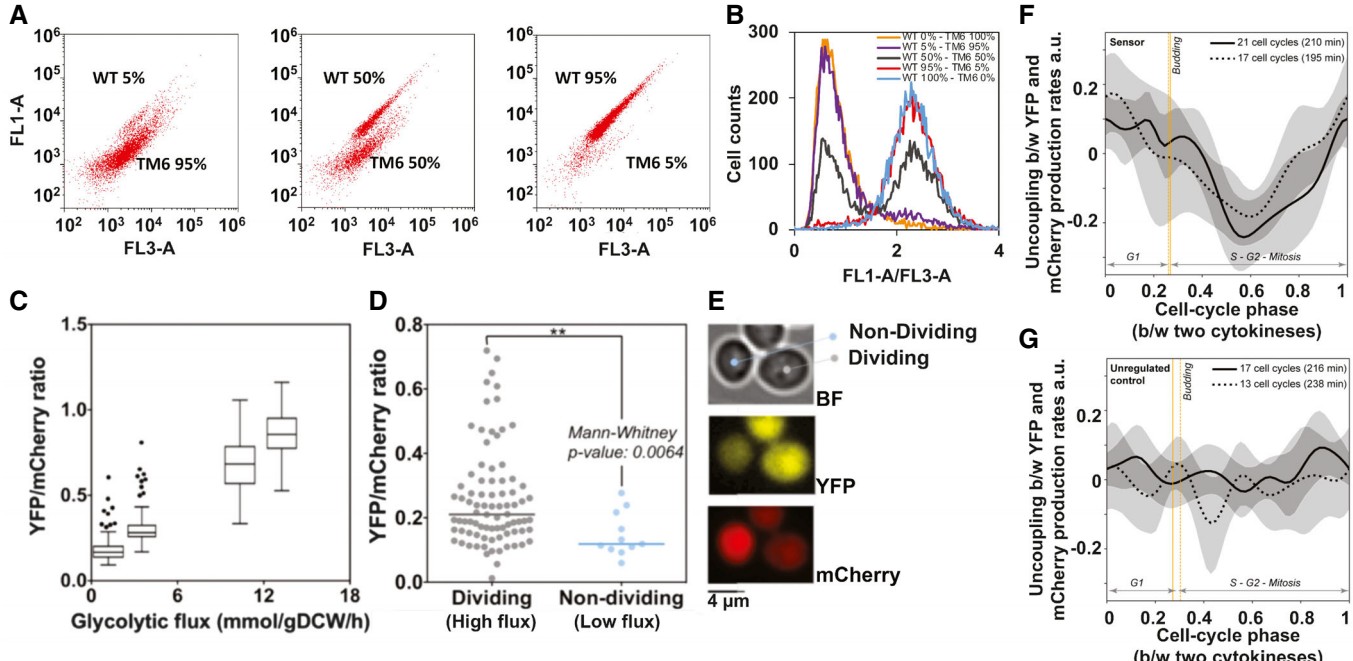

**Figure 7. The glycolytic flux sensor can measure glycolytic flux in individual cells.**

A   Subpopulations of WT and TM6 cells, grown separately and mixed in different fractions as indicated in percentages, can easily be distinguished by flow cytometry, FL1 YFP channel, FL3 mCherry channel.

B   Histogram of single-cell ratios of FL1/FL3 fluorescence intensities of mixed WT and TM6 populations analyzed by flow cytometry. Here, a subpopulation of minimally 5% can be distinguished.

C   Tukey boxplots showing the YFP/mCherry ratio of individual cells measured by microscopy as a function of glycolytic flux. At least 35 cells were analyzed in each condition. The glycolytic flux is here reported as the flux between the metabolites fructose 6-phosphate (F6P) and FBP. Glycolytic fluxes were estimated on the basis of physiological and metabolome data and a novel method to estimate intracellular fluxes (Niebel *et al*, 2019). The boxplot horizontal line indicates the median and the box extends from the 25th to 75th percentiles. Plotted points are outliers that are higher or lower than the upper and lower whiskers, respectively.

D   YFP/mCherry ratio measured by microscopy in co-existing dividing (high flux) versus non-dividing (low flux) isogenic TM6 cells on 10 g/l glucose. Each data point corresponds to data from a single cell.

E   Brightfield (BF), YFP, and mCherry microscopy images for a co-existing dividing (high flux) and a non-dividing (low flux) TM6 cell expressing the flux sensor in 10 g/l glucose minimal medium.

F, G   The production rates of YFP and mCherry are uncoupled during the cell cycle in the biosensor-expressing strain (F), which reflects the cell-cycle dynamics of intracellular FBP concentration and glycolytic flux. In a control strain, lacking CggR, the production rates of YFP and mCherry are coupled (G). The uncoupling was calculated for individual cell-cycle trajectories as the difference between the YFP and mCherry production rates normalized to have the same scale (see more details in Materials and methods; Appendix Fig S10). Each curve represents the mean across the indicated number of cell cycles in a replicate experiment. The corresponding shaded areas denote the 95% confidence intervals of the means (bootstrapping with 5,000 iterations). We smoothed the single-cell-cycle trajectories of YFP and mCherry signals as well as cell volume via the Gaussian process regression, and used these trajectories to derive the YFP and mCherry production rates, accounting for fluorescent-protein maturation in a first-order kinetics model. To align the cell-cycle trajectories and to calculate the phase, we used the array of three cell-cycle events $E$ {cytokinesis (cyt), budding, next cyt} as reference points. Specifically, we computed the average cell-cycle-relative timing for each of these events $\bar{\varphi}^e$ in the following way: $\forall e \in E \bar{\varphi}^e = \frac{1}{N}\sum_{cc=1}^{N} \frac{t_{cc}^e - t_{cc}^{cyt}}{t_{cc}^{next\,cyt} - t_{cc}^{cyt}}$, where $N$ is the number of cell cycles in the replicate of interest, and $t_{cc}^e$ is the time in minutes when the event $e$ happens in the cell cycle $cc$. The orange vertical lines denote $\bar{\varphi}^{budding}$ for both replicates. In the aligned cell cycles, we converted the time in minutes $t$ to the phase $\varphi_{cc}$ in the following way: $\varphi_{cc} = \left(\bar{\varphi}^{E[i+1]} - \bar{\varphi}^{E[i]}\right)\frac{t - t_{cc}^{E[i]}}{t_{cc}^{E[i+1]} - t_{cc}^{E[i]}} + \bar{\varphi}^{E[i]}$ for $t \in \left[t_{cc}^{E[i]}, t_{cc}^{E[i+1]}\right]$ if $E[i]$ = cyt or $t \in \left(t_{cc}^{E[i]}, t_{cc}^{E[i+1]}\right)$ if $E[i] \neq$ cyt, where $i$ is the index number of an event in the array $E$. The cell cycles used for the analysis had the duration in the interval between 150 and 300 min, with the mean duration presented in parentheses for each replicate experiment. The cells belonged to the TM6 strain and were cultivated on 20 g/l glucose in the microfluidic device.

Here, we found that in cells expressing the biosensor (i.e., CggR and the reporter plasmid) the YFP and mCherry production rates are uncoupled during the cell cycle, with YFP being produced faster relative to mCherry around cytokinesis and in G1, but markedly slower in the middle of S-G2-mitosis (Fig 7F, Appendix Fig S10A). In the strain without CggR (unregulated control), we observed no uncoupling between the YFP and mCherry production rates (Fig 7G, Appendix Fig S10B). Since YFP expression is controlled by FBP in the biosensor, this result showed that the FBP concentration, and thus glycolytic flux, change during the cell cycle, peaking around cytokinesis and in the G1

phase, inline with recent work (Litsios *et al*, 2019). This experiment demonstrates that the biosensor is also applicable to assess the dynamic behavior of metabolism in single cells.

## Discussion

Here, exploiting the flux-signaling metabolite fructose-1,6-bisphosphate and the bacterial transcription factor CggR, we developed a biosensor that allows to measure glycolytic flux in individual living

yeast cells, at least under glycolytic conditions. These engineering efforts, for which we used computational protein design, biochemical, proteome, and metabolome analyses, entailed (i) development of a synthetic yeast promoter regulated by the bacterial transcriptional factor CggR, (ii) engineering of the transcription factors' FBP-binding site toward increasing the sensor's dynamic range, and (iii) establishment of growth-independent CggR expression levels. Through single-cell flow cytometry and time-lapse fluorescence microscopy experiments, we demonstrated the applicability of the sensor to reveal differences in glycolytic flux between single cells.

Biosensor development based on transcription factors has recently seen rapid development (Rogers et al, 2016; Lehning et al, 2017; Liu et al, 2017; Carpenter et al, 2018). Yet, three aspects of our work are worth to be highlighted: As no endogenous FBP-binding transcription factor is known in yeast, we had to transfer the B. subtilis transcription factor CggR into yeast. However, unlike previous studies, which transplanted bacterial transcription factors into eukaryotes (Teo & Chang, 2015; Wang et al, 2016; Rantasalo et al, 2018), to ensure full orthogonality of the introduced sensing system to the host, we avoided the use of yeast-endogenous elements, such as DNA-binding domains or chimeric fusions of TF with transcriptional activation domains, and the use of a nuclear localization sequences. Instead, in our design, we build the promoter from scratch and exploited the natural mode of action of the TF also in the new host. Specifically, we used the ability of CggR to dimerize to allow for an effective repression mechanism also in yeast. Furthermore, and also in contrast to previous promoter engineering approaches, which often employed FACS-based screening approaches with large promoter libraries (Skjoedt et al, 2016), our approach was not a screening but a rational design approach. Our successful de novo engineering of a cis-regulatory element demonstrates that rational promoter development is possible when taking crucial factors into account, such as positioning and number of cis-regulatory elements, the transcription factor's mode of action, and the genomic context, i.e., nucleosome affinity.

A second important aspect in our biosensor development was that we made sure that the output of the sensor is not influenced by growth-dependent changes in the transcription factor's expression level, as its concentration also determines the synthesis rate of the gene product, and thus the output signal. In previous work, this point has mostly been ignored and TFs were typically "constitutively" expressed, although unregulated expression does not necessarily lead to constant expression level across different growth rates (Klumpp et al, 2009). Constant and condition-independent levels of the transcription factor are particularly important in light of a glycolytic flux sensor, which likely will be applied across growth conditions. To accomplish growth rate-independent expression levels, using quantitative proteomics, we found that the $P_{TEFmut7}$ promoter leads to more or less condition-independent levels of CggR, while two other tested constitutive promoters, i.e., $P_{CMV}$ and $P_{TEFmut2}$, showed strong growth-dependent expression levels. We hope that future development work toward transcription-dependent biosensors will also consider the expression level of the TF as an important element in the development of the sensor.

Another important aspect in our biosensor development was the optimization of the biosensor's dynamic range with regard to the sensed FBP levels and thus glycolytic fluxes. Here, we lowered the CggR–FBP-binding affinity to better cover the range of the intracellular FBP levels. Optimizing TF-effector sensitivity is not trivial, because transcription factors contain both an effector-binding domain and a DNA-binding domain, which should not be altered when engineering the former. Here, we applied a semi-rational design approach, supported by computationally guided protein design, to select mutants with lower FBP–CggR-binding activity and unaltered CggR–DNA-binding capacity. Demonstrating the challenge, only one mutation (R250A), out of a pool of 11 mutants, showed all desired features. Relevant for future engineering, only the mutations where the H-bond forming side chains were eliminated (R250A and T151V) resulted in the desired affinity loss. Furthermore, computational modeling with FoldX on the basis of available X-ray structures of both the normal and the activated conformation of the CggR effector-binding domain allowed to predict the instability or DNA-loss binding of some variants, indicating that computational stability predictions can successfully eliminate at least some dysfunctional mutants.

To construct the biosensor for glycolytic flux in yeast, we exploited the function of fructose-1,6-bisphosphate as a flux-signaling metabolite (Huberts et al, 2012) and we took advantage of the fact that FBP modulates the conformation of the B. subtilis transcription factor CggR (Doan & Aymerich, 2003). Can a similar approach be pursued to develop flux sensors also for other metabolic pathways? This seems possible: On the basis of metabolite dynamics assessed across various nutrient conditions and known metabolite–protein and metabolite–RNA interactions, we recently compiled a list of several other candidates of flux-signaling metabolites (Litsios et al, 2018), which includes citrate, alpha-ketoglutarate, phosphoenolpyruvate, pyruvate, and succinate. Exploiting respective metabolite-binding transcription factors and engineering biosensors for these metabolites should yield flux sensors also for other pathways. For instance, for citrate, whose concentration correlates with the cellular nitrogen flux (Fendt et al, 2013), the transcriptional activator CitI from lactic acid bacteria (Martin et al, 2005) would be an excellent starting point. Thus, as flux-signaling metabolites also exist for other metabolic pathways (Litsios et al, 2018), and transcription factors exist for many of these metabolites (Reznik et al, 2017), it should be possible to develop flux biosensors also for other metabolic pathways.

We envision that our glycolytic flux biosensor, applicable in single living yeast cells, will find applications in fundamental research with Saccharomyces cerevisiae, i.e., to address the daunting emergence of metabolic heterogeneity as occurring during replicative aging (Leupold et al, 2019) or cellular growth (van Heerden et al, 2014; Kiviet et al, 2014; Thomas et al, 2018) or to investigate metabolic dynamics during the cell cycle (Papagiannakis et al, 2016; Litsios et al, 2019). Furthermore, we expect that the biosensor will also have value for applied research. Metabolic heterogeneity is a significant problem in industrial fermentations, especially those with cell recycling as applied in beer brewing and bioethanol production (Stewart et al, 2013; Aranda et al, 2019; Wang et al, 2019), where physiological and genetic changes can cause losses in fermentation performance (Powell et al, 2003). In such large-scale yeast applications, our glycolytic flux sensor will provide a tool to study how and why metabolic subpopulations with high or low glycolytic flux phenotypes emerge. Beyond, we expect that the biosensor will find its application also as a screening tool in metabolic engineering efforts, for instance to screen for highly productive phenotypes, rather than just for selecting on growth.

# Materials and Methods

## Reagents and Tools table

| Reagent/Resource | Reference or source | Identifier or catalog number |
|---|---|---|
| **Experimental models** | | |
| *Saccharomyces cerevisiae* wild-type (WT) ura⁻ | (Elbing *et al*, 2004) | N/A |
| *Saccharomyces cerevisiae* TM6 | (Elbing *et al*, 2004) | N/A |
| **Recombinant DNA** | | |
| pET100-CggR-Sc | This study | N/A |
| HO-poly-KanMX4-HO | ATCC | 87804 |
| pUG66 | EUROSCARF | P30116 |
| pCM149 | EUROSCARF | P30344 |
| p416-loxP-KmR-TEFmut2-yECitrine | (Nevoigt *et al*, 2006) | N/A |
| p416-loxP-KmR-TEFmut7-yECitrine | (Nevoigt *et al*, 2006) | N/A |
| p416-loxP-KmR-TEFmut6-yECitrine | (Nevoigt *et al*, 2006) | N/A |
| p416-loxP-KmR-TEFmut8-yECitrine | (Nevoigt *et al*, 2006) | N/A |
| pHO_pCMV_CggR_ble | This study | N/A |
| pHO_pTEFmut2_CggR_ble | This study | N/A |
| pHO_pTEFmut7_CggR_ble | This study | Addgene 124584 |
| pHO_pTEFmut2_CggR_R250A_ble | This study | N/A |
| pHO_pTEFmut7_CggR_R250A_ble | This study | Addgene 124585 |
| pBS35 | Yeast Resource Center | pBS35 |
| pWHE601 | Beatrix Suess Lab | N/A |
| pYCplac33 | ATCC | 87586 |
| pTEF6-7 | This study | Addgene 124583 |
| pCggRO reporter | This study | Addgene 124582 |
| Additional plasmid information | This study | Appendix Table S5 |
| **Oligonucleotides and other sequence-based reagents** | | |
| PCR primers | This study | Appendix Tables S6, S7 and S9 |
| **Chemicals, enzymes, and other reagents** | | |
| Phusion® High-Fidelity DNA Polymerase | New England Biolabs | M0530 |
| Antarctic phosphatase | New England Biolabs | M0289 |
| T4 DNA ligase | New England Biolabs | M0202 |
| Trypsin spectrometry grade | Promega | V5280 |
| DpnI | New England Biolabs | R0176 |
| SYPRO® Orange Protein Gel Stain | Sigma-Aldrich | S5692 |
| Alexa Fluor 647 | NanoTemper Technologies | NHS RED Kit |
| Kanamycin | Sigma-Aldrich | 60615 |
| Gel and PCR Clean-up kit | Macherey-Nagel | 740609 |
| Gibson assembly kit | New England Biolabs | E5510S |
| Pierce™ BCA Protein Assay | Thermo Fisher | 23225 |
| Nucleospin plasmid purification kit | Macherey-Nagel | 740588 |
| HPLC column | Agilent | Hi-Plex H column for carbohydrates |
| UPLC Column HSS T3 | Waters | Waters Acquity UPLC HSS T3 |
| Verduyn minimal media | (Verduyn *et al*, 1992) | N/A |

**Reagents and Tools table** (continued)

| Reagent/Resource | Reference or source | Identifier or catalog number |
|---|---|---|
| **Software** | | |
| Agilent Open Lab CDS software | https://www.agilent.com/en/products/software-informatics/chromatography-data-systems/openlab-cds | N/A |
| gPROMS Model Builder v.4.0 | gPROMS Model Builder v.4.0 | N/A |
| Software for Accuri flow cytometer | CFlow Plus Analysis | N/A |
| | Kaluza Analysis Software | N/A |
| R version 3.4.0, RStudio version 1.0.143 | https://www.R-project.org/,https://rstudio.com/ | N/A |
| Python version 3.6.2 | https://www.python.org/ | N/A |
| BudJ | (Ferrezuelo *et al*, 2012) | N/A |
| ImageJ | (Abràmoff *et al*, 2004) | N/A |
| General algebraic modeling system (GAMS) | https://www.gams.com/ | |
| Thermodynamic constraint-based metabolic model | (Niebel *et al*, 2019) | N/A |
| optGpSampler | (Megchelenbrink *et al*, 2014) | N/A |
| Progenesis QI | http://www.nonlinear.com/progenesis/qi/ | N/A |
| SafeQuant R script | (Ahrné *et al*, 2016) | N/A |
| FoldX | (Guerois *et al*, 2002) | N/A |
| GraphPad Prism 8 | https://www.graphpad.com | N/A |
| SnapGene | https://www.snapgene.com | N/A |
| ApE plasmid editor | http://jorgensen.biology.utah.edu/wayned/ape/ | N/A |
| **Other** | | |
| 1290 LC HPLC system | Agilent | N/A |
| Dionex Ultimate 3000 RS UHPLC | Dionex | N/A |
| MDS Sciex API365 tandem mass spectrometer | Ionics | N/A |
| Turbo-Ion spray source | MDS Sciex | N/A |
| BD Accuri™ C6 flow cytometer | BD Biosciences | N/A |
| LTQ-Orbitrap Elite mass spectrometer | Thermo Fisher | N/A |
| CFX96 Real-Time System combined with C1000 Touch Thermal Cycler | Bio-Rad | N/A |
| Typhoon 9400 | Amersham Biosciences | N/A |
| Microfluidic chip | (Lee *et al*, 2012; Huberts *et al*, 2013) | N/A |
| Eclipse Ti-E inverted fluorescence microscope | Nikon | N/A |
| pE2 LED-based excitation system | CoolLED | N/A |
| Andor 897 Ultra EX2 EM-CCD camera | Andor | N/A |

**Methods and Protocols**

*Generation and cloning of the CggR cis-regulatory element and reporter plasmid*

The overall architecture of the four promoter elements is outlined in the main text. The total DNA fragment size of all promoter element was 562 bp. This DNA fragment included two ends complementary to the reporter plasmid to allow for the Gibson assembly of the reporter plasmid and the synthetic promoter. The complementary flanking sites of the promoter element had a size of 100 bp at the 5′ end and 145 bp at the 3′ end.

The first promoter elements were composed of the wild-type sequence of the *CYC1* core promoter (core promoter of *CYC1*). In the second promoter construct, the cggR-binding elements were introduced by replacing the sequences of the *CYC1* promoter at the insertion positions (core-*CYC1* with cggRO). In addition, two synthetic

promoter sequences were generated using a computational method to minimize nucleosome affinity (Curran *et al*, 2014). The initial sequence of the CggR cis-regulatory element was generated with a random sequence used as the starting point for further sequence optimization (syn CggRO promoter V1). The functional elements of this synthetic construct, i.e., the CggR operator site, the TATA boxes, and the 5′UTR, were excluded from the sampling procedure and thus remained conserved. The algorithm was run for 75 rounds. The version 38 (syn CggRO promoter V38) of the optimized CggR cis-regulatory element was selected since it showed the lowest affinity to nucleosomes and it was still feasible to be synthesized. DNA synthesis was performed with a STRING™ DNA fragment (GenArt™, Thermo Fisher Scientific, MA, USA) and directly used for the assembly of the reporter plasmid. The functional DNA sequences of the CggR cis-regulatory element and their distance from the 5′ end of the synthesized fragment are listed in Appendix Table S1, and the synthesized sequences are given in Appendix Table S2.

To account for extrinsic cell-to-cell variation in the state of the gene and protein expression machinery, a constitutively expressed mCherry reporter was inserted into the low copy p416-loxP-KmR-TEFmut6-yECitrine centromeric plasmid (Nevoigt *et al*, 2006). The mCherry cassette included the mCherry ORF, the constitutive *TEF1* promoter mutant 8 ($P_{TEFmut8}$), and the *ADH1* terminator, amplified from the plasmids pBS35, p416-loxP-KmR-TEFmut8-yECitrine, and pWHE601 (Appendix Table S5), respectively, with the primers listed in Appendix Table S6. The three DNA fragments, *i.e.,* $P_{TEFmut8}$, mCherry, and *ADH1* terminator, were assembled by PCR using the Phusion® High-Fidelity DNA Polymerase (New England Biolabs, MA, USA) and the mCherry_KpnI_fw and mCherry_KpnI_rv primers (Appendix Table S6). The resulting 1.4-bp DNA fragment was purified, digested with KpnI, and purified again with PCR Clean-up kit (Macherey-Nagel, Germany). The p416-loxP-KmR-TEFmut6-yECitrine plasmid was linearized with KpnI and dephosphorylated with Antarctic Phosphatase (New England Biolabs, MA, USA), and ligated with the mCherry expression cassette by T4 DNA ligase (New England Biolabs, MA, USA). The ligation assay was transformed into chemical competent *E. coli* DH5alpha cells. Clone screening was performed by sequencing the extracted plasmids with the primers listed in Appendix Table S6.

To construct a plasmid carrying the regulated CggR promoter, we used the p416-loxP-KmR-TEFmut6-yECitrine with the inserted mCherry cassette (pTEF6-7) (Appendix Table S5) and replaced the $P_{TEFmut6}$ promoter by the CggR cis-regulatory element using Gibson assembly. The backbone of the pTEF6-7 plasmid was divided into three fragments, which were amplified by PCR using the primers listed in Appendix Table S6. The three backbone fragments were combined together with the synthesized CggR cis-regulatory element using the NEB Gibson assembly kit (New England Biolabs, MA, US) according to the manufacturer instructions. 5 μl of reaction mix was transformed into chemical competent *E. coli* cells. Clone screening was performed to isolate the correct assembled pCggRO-reporter plasmid. The plasmid sequence was verified by Sanger sequencing of the extracted plasmids with the primers listed in Appendix Table S6.

### Cloning of the CggR regulator and its variants and promoters

The open reading frame of the transcription regulator CggR of *B. subtilis* was codon-optimized for expression in *S. cerevisiae*. A

His$_6$ and an Xpress epitope tag were added at the N-terminus of the protein. The CggR sequence was assembled from synthetic oligonucleotides (Thermo Fisher Scientific GeneArt AG, Germany), and the ORF was subcloned in the pET100/D-TOPO express cloning vector. Next, a Gibson assembly (New England Biolabs, MA, USA) was carried out to generate an expression cassette of the *cggR* gene for further integration in the HO locus of *S. cerevisiae* genome. To allow for integration, the *cggR* gene was cloned into the integrative plasmid HO-poly-KanMX4-HO, where the KanMX4 resistance marker was replaced by the *ble* resistance gene. To select for the transformants, we either used ura⁻ auxotrophy or phleomycin, for WT or TM6, respectively.

Promoter and terminator of the *cmv* gene were amplified from the pCMV149 with a 5′ and 3′ overhang of a homologous sequence to the codon-optimized *cggR* gene. The three DNA fragments, *i.e.,* the *cggR* gene, the CMV promoter ($P_{CMV}$), and terminator, were combined together by PCR using the Phusion® High-Fidelity DNA Polymerase (New England Biolabs, MA, USA). The resulting fragment was 2.2 kbp was gel-purified. A Gibson assembly was carried out to link all DNA fragments and assemble the final integrative pHO_pCMV_CggR_ble plasmid using the NEB Gibson assembly kit (New England Biolabs, MA, US). 5 μl of reaction mix was transformed into chemical competent *E. coli* DH5alpha cells. Clone screening was performed by sequencing the extracted plasmids. All the plasmids and primers used to generate the integrative plasmids are listed in Appendix Tables S5 and S6, respectively. Additionally, the constitutive promoters TEF mutant 2 ($P_{TEFmut2}$) and mutant 7 ($P_{TEFmut7}$) (Nevoigt *et al*, 2006) were also cloned for testing the effect of different expression CggR levels on the biosensor output. $P_{TEFmut2}$ and $P_{TEFmut7}$ were amplified from p416-loxP-KmR-TEFmut2-yECitrine and p416-loxP-KmR-TEFmut7-yECitrine, respectively, and used to replace the $P_{CMV}$ in the pHO_pCMV_CggR_ble plasmid.

To address the effect of a lower $K_D$ toward FBP, the CggR mutant variant R250A was inserted in the above generated plasmids (replacing the CggR wild type) using the NEB Gibson assembly kit (New England Biolabs, MA, US). 5 μl of reaction mix was transformed into chemical competent *E. coli* cells. Clone screening was performed by sequencing the extracted plasmids. The set of primers used for each integrative plasmid generation is listed in Appendix Table S7. The strains generated and used throughout this work are listed in Appendix Table S8.

### Cultivation and experimental sampling

All strains were cultivated in 500-ml Erlenmeyer shake flask containing 50 ml of minimal medium (Verduyn *et al*, 1992) inoculated with exponentially growing yeast cells to an initial OD$_{600}$ of 0.1–0.2 (ca. 1–2 × 10$^7$ cells). To adapt to the carbon source and to ensure metabolic steady state, two pre-culturing steps were carried out prior to the main culture. The inoculum was prepared in the identical minimal medium. All cultivations were performed at 30°C, and cultures were continuously shaken at 300 rpm. The medium was buffered at pH 5 with 10 mM KH phthalate. Cells were cultured in different carbon sources that would generate distinct glycolytic fluxes and, as a consequence, FBP levels. Specifically, WT cells were grown in minimal medium containing 10 g/l of glucose, galactose, or pyruvate and TM6 cells were cultured in minimal medium containing 10 g/l of maltose, glucose, or pyruvate.

Cell counts were performed by flow cytometry (BD Accuri™ C6 Flow Cytometer, BD Biosciences, CA, USA) every hour for glucose, galactose, and maltose. YFP and mCherry expression was assessed by measuring the fluorescence along culture time using flow cytometry through the FL1-A and FL3-A filters, respectively. Autofluorescence was assessed by measuring the fluorescence in cells containing the centromeric yeast plasmid YCplac33 (Gietz & Akio, 1988) as a control using the FL1 and FL3 filters.

To determine the production and consumption rates of metabolites during the cultivation, supernatant samples were taken every hour from the cultivations. 0.3 ml of the broth sample was centrifuged at 13 rpm for 2 min to separate the cells from the supernatant. The supernatant was transferred to filter columns (SpinX, pore size: 0.22 μm), spun shortly, and stored at −20°C until HPLC analyses were performed. At the end, the yeast dry mass was determined by filtering a certain volume of culture through pre-weighed nitrocellulose filters with a pore size of 0.2 μm. Filters were washed once with distilled water and kept at 80°C for 2 days. Afterward, they were weighed again. The cell dry mass at every measurement point was re-calculated from cell count and the dry mass cell count/ratio using the dry mass cell count/ratio obtained at the end of the fermentation.

### Quantification of physiological parameters

Glucose, pyruvate, glycerol, acetate, and ethanol concentrations in the cultivation supernatant were determined by HPLC (Agilent, 1290 LC HPLC system) using a Hi-Plex H column and 5 mM $H_2SO_4$ as eluent at a constant flow rate of 0.6 ml/min. The column temperature was kept constant at 60°C. A volume of 10 μl of standards and samples was injected for analysis. Substrate concentrations were detected with refractive index and UV (constant wavelength of 210 nm) detection. The chromatogram integration was done with Agilent Open Lab CDS software. Substrate and metabolite concentration were calibrated prior to the analysis of the fermentation samples using HPLC standards, which included all metabolites, relevant for the various conditions. The external standards covered the metabolites' concentration range which was observed from the start until the end of the fermentation.

Carbon uptake rate calculations were performed using the time-course data of the exponentially growing cultures of the different strains and carbon sources. From at least three independent biological replicates, the extracellular rates were estimated from measured concentration–time courses, e.g., glucose, ethanol, acetate, glycerol, pyruvate, and biomass, of the batch cultivation. Extracellular rates were estimated by fitting the concentration–time courses to a mathematical model assuming exponential growth and constant yields in the culture. The regression and parameter estimation were implemented in gPROMS Model Builder v.4.0 (Process Systems Enterprise Ltd.).

### Quantification of intracellular metabolite levels

A sample of $3 \times 10^7$ cells was taken and immediately quenched in 10 ml methanol, which was beforehand cooled down to −40°C. The cells were separated from the organic solvent by centrifugation (5 min, 21,000 $g$, 4°C), washed with 2 ml methanol, separated again by centrifugation, and stored at −80°C. For the following analysis, the cell pellet was resuspended in extraction buffer (methanol, acetonitrile, and water, 4:4:2 v/v/v supplemented with 0.1 M formic acid) and an internal standard of $^{13}$C-labeled metabolites was added

to the extraction. This standard was obtained and quantified from exponentially grown cell cultures prior to the experiment. The extraction was agitated for 10 min at room temperature and thereafter centrifuged at maximum speed. The supernatant was transferred to a new vial and the cell pellet resuspended in extraction buffer and the extraction procedure was repeated a second time. The supernatants from both steps were combined and centrifuged for 45 min at 4°C and 21,000 $g$ to remove any remaining non-soluble parts. Thereafter, the supernatant was vacuum-dried at 45°C for approximately 1.5 h and subsequently re-dissolved in 200 μl water.

The extracted intracellular metabolites were identified and quantified using a UHPLC-MS/MS system as done previously (Radzikowski et al, 2016). Specifically, the chromatographic separation was performed on a Dionex Ultimate 3000 RS UHPLC (Dionex, Germering, Germany) equipped with a Waters Acquity UPLC HSS T3 ion pair column with pre-column (dimensions: 150 × 2.1 mm, particle size: 3 μm; Waters, Milford, MA, USA). The injection volume was 10 μl, and the samples were permanently cooled at 4°C. A binary solvent gradient was employed (0 min: 100% A; 5 min: 100% A; 10 min: 98% A; 11 min: 91% A; 16 min: 91% A; 18 min: 75% A; 22 min: 75% A; 22 min: 0% A; 26 min: 0% A; 26 min: 100% A; 30 min: 100% A) at a flow rate of 0.35 ml/min, where solvent A was composed of 5% (v/v) methanol in water supplemented with 10 mM tributylamine, 15 mM acetic acid, and 1 mM 3,5-heptanedione and isopropanol as solvent B. The detection was done using multiple reaction monitoring (MRM) on a MDS Sciex API365 tandem mass spectrometer upgraded to EP10 + (Ionics, Bolton, Ontario, Canada) and equipped with a Turbo-Ion spray source (MDS Sciex, Nieuwerkerk aan den Ijssel, Netherlands) with the following source parameter: NEB (nebulizing gas, N2): 12 a.u., CUR (curtain gas, N2): 12 a.u., CAD (collision activated dissociation gas): 4 a.u., IS (ion spray voltage): −4,500 V, and TEM (temperature): 500°C.

The amounts of metabolites determined in each sample were converted into intracellular concentrations, using the determined number of cells in the sample and the respective cell volume. The cell volume was determined by taking an image of mid-exponential cells of wild type (WT) and TM6 in various conditions. The cells were placed on a microscopic slide. Approximately 200 cells per condition and replicate were evaluated on several positions/images of the microscopic slide. The cell volume was estimated using BudJ plugin of ImageJ (Ferrezuelo et al, 2012).

### Determination of intracellular fluxes

To estimate the glycolytic flux for the two strains and the different substrate conditions, we used a thermodynamic constraint-based metabolic model, and a new approach for metabolic flux analysis (Niebel et al, 2019). The model and constraints were based on what we previously used, but the network was extended by reactions describing the uptake and metabolization of galactose and maltose (Appendix Tables S10 and S11). The model was fitted to experimental data, which here comprised of intracellular metabolite concentrations, extracellular fluxes (as measured for the different conditions and strain), and standard Gibbs free energies of reaction ($\Delta_r G°$), determined from the component contribution method (Noor et al, 2013). The experimental data of all six conditions are given in Appendix Tables S12 and S13. The fitting was done as previously, i.e., jointly for all conditions to identify one condition-dependent set of $\Delta_r G^O$, but without regularization. The

result of the regression analysis and the goodness of fit are presented in Appendix Fig S9. Next, we performed a flux variability analysis to determine the limits of the solution space of each flux for all growth conditions and the two strains, for which the fitted values of the measured extracellular fluxes were allowed to vary ± 2.95 standard deviations. Within those limits, 1000 flux distributions were sampled for each condition with optGpSampler (Megchelenbrink *et al*, 2014) using linear approximations for the non-linear thermodynamic constraints, as we did previously (Niebel *et al*, 2019). The mean and standard error of each metabolic flux at each condition were determined from this sample population.

### Quantification of CggR and mCherry levels

Intracellular CggR and mCherry levels were quantified in CMV, $P_{TEFmut2}$, and $P_{TEFmut7}$_CggR yeast strains. WT and TM6 cells were grown using the same scheme as described above. Cell counts were monitored by flow cytometry (BD Accuri™ C6 flow cytometer (BD Biosciences, CA, USA), and cells were harvested once they reached a concentration of $10^7$ cells/ml. To do so, 10 ml of cells was pelleted (ca. $10^8$ cells per sample), washed twice with ice-cold PBS (0.1 mM $Na_2HPO_4$, 0.018 mM $KH_2PO_4$, 1.37 mM NaCl, 0.027 mM KCl), and centrifuged at 10,000× *g*, at 4°C for 10 min. The cell pellet was frozen in liquid nitrogen and stored at −80°C until further analysis. For each strain, three biological replicates and three technical replicates were taken.

The cell pellet was reconstituted in 40 μl 2% sodium deoxycholate (SDC); 10 mM TCEP; and 100 mM ammonium bicarbonate and sonicated two times for 10 s using a UP200St with VialTweeter (Hielscher Ultrasonics GmbH, Germany). Heat treatment was performed for 10 min at 95°C. After cooling, the protein concentration was determined by BCA assay for each sample (Thermo Fisher Scientific, MA, USA). Sample aliquots containing 100 μg protein were used for the following steps. Alkylation was performed by adding iodoacetamide to a final concentration of 15 mM and incubation for 45 min at RT, in the dark. The samples were diluted to 1% SDC using 100 mM ammonium bicarbonate and mass spectrometry grade trypsin (Promega, WI, USA) was added at a ratio of 1:50 (μg trypsin:μg protein) and samples were incubated overnight at 37°C, 400 rpm. The reaction was stopped by adding trifluoroacetic acid to a final concentration of 1%. Sample clean-up by solid-phase extraction was performed with Pierce® C18 tips (Thermo Fisher Scientific, MA, USA) according to the manufacturer instructions. The eluate fraction was dried under vacuum and reconstituted with 20 μl of a mixture of 2% acetonitrile and 0.1% formic acid.

1 μg of peptides of each sample was subjected to LC-MS analysis using a dual pressure LTQ-Orbitrap Elite mass spectrometer connected to an electrospray ion source (Thermo Fisher Scientific, MA, USA) as described recently (Ahrné *et al*, 2016) with a few modifications. In brief, peptide separation was carried out using an EASY nLC-1000 system (Thermo Fisher Scientific, MA, USA) equipped with a RP-HPLC column (75 μm × 30 cm) packed in-house with C18 resin (ReproSil-Pur C18–AQ, 1.9 μm resin; Dr. Maisch GmbH, Ammerbuch-Entringen, Germany) using a linear gradient from 95% solvent A (0.15% formic acid, 2% acetonitrile) and 5% solvent B (98% acetonitrile, 0.15% formic acid) to 28% solvent B over 100 min and to 45% B over 20 min at a flow rate of 0.2 μl/min. The data acquisition mode was set to obtain one high-resolution MS scan in the FT part of the mass spectrometer at a resolution of 120,000 full width at half-maximum (at *m/z* 400) followed by MS/MS scans in the linear ion trap of the 20 most intense ions using rapid scan speed. The charged state screening modus was enabled to exclude unassigned and singly charged ions, and the dynamic exclusion duration was set to 60 s. The ion accumulation time was set to 300 ms (MS) and 25 ms (MS/MS).

For label-free quantification, the generated raw files were imported into the Progenesis QI software (Nonlinear Dynamics (Waters), Version 2.0) and analyzed using the default parameter settings. MS/MS data were exported directly from Progenesis QI in mgf format and searched against a decoy database the forward and reverse sequences of the predicted proteome from *Saccharomyces cerevisiae* (strain ATCC 204508/S288c, UniProt, download date: 15/12/2016) including common contaminants, such as keratins and CggR from *Bacillus subtilis* (strain 168, total of 14,248 entries) using MASCOT (version 2.4.1). The search criteria were set as follows: Full tryptic specificity was required (cleavage after lysine or arginine residues); three missed cleavages were allowed; and carbamidomethylation (C) was set as fixed modification and oxidation (M) as variable modification. The mass tolerance was set to 10 ppm for precursor ions and 0.6 Da for fragment ions. Results from the database search were imported into Progenesis QI, and the final peptide measurement list containing the peak areas of all identified peptides, respectively, was exported. This list was further processed and statically analyzed using our in-house developed SafeQuant R script (Ahrné *et al*, 2016). The peptide and protein false discovery rate (FDR) were set to 1% using the number of reverse hits in the dataset.

The CggR intracellular abundance is the cell protein content of CggR calculated using the iBAQ approach (Ahrné *et al*, 2013). The relative abundance of CggR was calculated by normalizing the cell protein content of CggR to the cell protein content of the CggR measured on glucose and using same promoter.

### *In silico* design of a CggR mutant library with altered affinity for FBP

FoldX was used to predict the stability and structure of CggR mutants (Guerois *et al*, 2002). FoldX evaluates the change in folding energy ($\Delta\Delta G^{fold}$) due to a point mutation. This $\Delta\Delta G^{fold}$ equals the Gibbs folding energy ($\Delta G^{fold}$) of the mutant minus the Gibbs folding energy of the wild-type protein. The FoldX settings were as default for the calculation of the stability effects of point mutations, with averaging of five independent predictions per variant (Wijma *et al*, 2018). In the dimeric CggR structure (PDB id: 3BXF), two conformations occur because the binding of FBP to only one of the subunits causes a local conformational change (Rezáčová *et al*, 2008). To model the effect of mutations on the equilibrium between those two conformations, two dimer structures were generated, in which the same conformation occurs in both subunits. This was done by duplicating one subunit, superimposing the resulting copy on the coordinates of the other subunit by a least-squares fit, and then eliminating the original subunit at that position. Suitable residues to mutate in the FBP-binding site were identified by visual inspection using Yasara (Krieger & Vriend, 2014). From a total of 18 mutations predicted, 6 were selected to be tested *in vitro* (Table 1) based on the desired mild impact on the dissociation constant ($K_d$) of CggR.

### Site-directed mutagenesis of CggR

For the generation of the CggR mutants with decreased FBP-binding affinity, a site-directed mutagenesis approach based on PCR was performed. The set of primers containing the mutation of interest are listed in Appendix Table S9. All the PCRs were performed according to the Phusion® High-Fidelity DNA Polymerase (New England Biolabs, MA, USA) protocol using 0.05 Units of Phusion and 100 μM dNTPs in a total volume of 25 μl.

To eliminate contamination of the original template DNA, a DpnI digestion was performed by adding 1 μl of DpnI to the PCR mix, followed by an overnight incubation at 37°C. 9 μl of the DpnI digestion product was then transformed into chemical competent *E. coli* DH5alpha cells. Plasmid DNA extraction was done with the nucleospin plasmid purification kit (Macherey-Nagel, Germany). The confirmation of the mutated CggR variants was performed by sequencing of the extracted plasmids.

### CggR protein expression and purification

CggR wild type and the generated mutants were cloned in pET100/D-TOPO (Thermo Fisher Scientific; Waltham, MA), with an N-terminal $His_6$ tag and expressed in *E. coli*. All constructs were verified by sequencing. For protein production, a single colony was used to inoculate 50 ml LB containing 100 μg/ml ampicillin, and the culture was grown at 37°C overnight. This culture was diluted to an optical density ($OD_{600}$) of 0.05 in a final volume of 2 l. Protein expression was induced at $OD_{600}$ 0.5 by addition of 10 μM IPTG, and cells were kept at 30°C and 180 rpm for four more hours. Cells were harvested by centrifugation at $6,675 \times g$ at 4°C for 20 min and washed once with 30 ml of 50 mM Tris–HCL buffer (pH 7.2). Cells were pelleted at $3,000 \times g$ at 4°C for 40 min, frozen in liquid nitrogen, and stored at −80°C until further use.

For protein purification, cell pellets were thawed on ice and resuspended with 10 ml of icecold lysis buffer (50 mM $KH_2PO4$, 300 mM NaCl, 1 mM EDTA, pH 7.5) per gram of cell pellet, and lysed by high-pressure disruption (Constant Cell Disruption System, Ltd, UK) in one passage at 25 Kpsi at 4°C. Prior to lysate centrifugation, 1 mM of PMSF, 20 mM $MgCl_2$, and 10 μg/ml of DNase (Sigma-Aldrich, MO, USA) were added. Cell debris was removed by centrifugation at $35,200 \times g$ at 4°C for 20 min. The cleared lysate was incubated with 0.5 ml of a nickel sepharose resin (GE Healthcare, Little Chalfont, UK), pre-equilibrated with 50 mM of $KH_2PO_4$ buffer (pH 7.5), and incubated at 4°C in batch mode overnight. The nickel sepharose-lysate suspension was poured onto a 10-ml disposable column (Bio-Rad), and the settled resin washed with 20 column volumes of a first wash solution (50 mM $KH_2PO_4$, 300 mM NaCl, 60 mM imidazole, pH 7.5), followed by 20 column volumes of a second wash solution (50 mM $KH_2PO_4$, 300 mM NaCl, 50 mM L-histidine, pH 7.5). Protein elution was performed with 9 ml of elution buffer (50 mM $KH_2PO_4$, 300 mM NaCl, 235 mM L-histidine, pH 7.5).

Protein concentration was measured by absorbance at 280 nm and extinction coefficient of 4.2. Protein purity and integrity were evaluated by running the samples in a 10% SDS–PAGE gel using protein concentrations of 0.1 mg/ml. A buffer exchange (50 mM $KH_2PO_4$, 300 mM NaCl, pH 7.5) was performed on the fractions with concentrations above 1 mg/ml and purity above 95%. The protein purified stocks were stored at 4°C until needed.

### Thermal shift assays

A sample mixture of 25 μl final volume containing 5 μl of 5× SYPRO Orange (Molecular Probes; Eugene, OR), 0.2 mg of the purified CggR, and different concentrations of the FBP metabolite (0; 0.01; 0.1; 0.5; 0.75; 1; 2.5; 10; 20; and 36 mM) was prepared on ice. Control experiments were performed to test the effect of the counter ion NaCl present in FBP salt solutions. Here, the NaCl was added instead of FBP to a final concentration threefold higher than the FBP itself (0; 0.03; 0.3; 1.5; 2.25; 3; 7.5; 30; 60; and 108 mM).

Sample mixtures were transferred into 96-well PCR plates (Bio-Rad, CA, USA), sealed with Optical-Quality Sealing Tape (Bio-Rad, CA, USA), and analyzed in a CFX96 Real-Time System combined with C1000 Touch Thermal Cycler (Bio-Rad, CA, USA). Analysis consisted of a single heating cycle from 20 to 99°C with increments of 0.5°C steps, followed by fluorescence intensity monitoring with a charge-coupled device camera. The wavelengths for excitation and emission were 490 and 575 nm, respectively. The melting temperature ($T_m$) was automatically calculated by the control software and corresponded to the local maximum of the first derivative of measured fluorescence versus temperature.

The $K_D$ of the wild type and mutant CggR variants was calculated by fitting the $T_m$ data into a simple cooperative model using the GraphPad software.

### Electrophoresis mobility shift assay

Fluorescently labeled DNA fragments were generated by hybridization of single-stranded forward (labeled with Alexa Fluor 647 at the 5′ end) and reverse (unlabeled) oligonucleotides containing the CggR operator sites. The hybridization protocol and respective hybridization efficiency control were performed as described elsewhere (Bley Folly *et al*, 2018).

Hybridized labeled DNA fragments (final concentration 35 nM) were incubated 20 min at room temperature with 2 or 4 μM of the purified wild type or mutant CggR variants, in the presence of different concentrations of FBP (0; 0.5; 1; 2.5; 5; 10; and 20 mM) in a final volume of 25 μl in binding buffer (10 mM $Na_3PO_4$ pH 7.8, 100 mM NaCl, 1 mM EDTA, 1 mM DTT, 5% glycerol) including 1 μg of salmon sperm DNA. The sample mixtures were loaded in a 5% native polyacrylamide gel and run in native conditions in the dark with constant voltage (200V) at 4°C for 2 h.

Fluorescence was imaged using the Typhoon 9400 (Amersham Biosciences, UK) with the excitation set to 650 nm and the emission set to 655 nm. The background-subtracted total intensities of the protein–DNA complex and the free-DNA bands were assessed using ImageJ (Abràmoff *et al*, 2004), and bound DNA/total DNA was calculated by dividing the intensity of the protein–DNA complex band by the total DNA (i.e., the sum of the signal from the protein–DNA complex band plus the one from free DNA).

### Time-lapse microscopic analyses

For testing the glycolytic flux-sensor output with microscopy, a microfluidic device (Lee *et al*, 2012; Huberts *et al*, 2013) was used, which was loaded with cells at log phase growing in glucose, maltose, or galactose. Monitoring of cells took place using an inverted fluorescence microscope (Eclipse Ti-E; Nikon). During the experiment, a custom-made microscope incubator (Life Imaging Services GmbH) retained the temperature constant at 30°C, and cells

were continuously supplied with fresh medium. For illumination, an LED-based excitation system (pE2; CoolLED) was used, and images were recorded with an Andor 897 Ultra EX2 EM-CCD camera using a CFI Plan Apo VC 60× Oil (NA = 1.4; Nikon) objective or a CFI Super Fluor 100XS Oil (NA = 0.50–1.30; Nikon) objective (in the experiments applying the sensor and unregulated control to the cell-cycle context). 300-ms exposure time and 50% light intensity were used for YFP (500-nm excitation using a 520/20-nm excitation filter and a 515-nm beam splitter, 535/30-nm emission filter, EM gain 25), and 200-ms exposure time and 25% light intensity were used for mCherry measurements (565-nm excitation using a 562/40-nm excitation filter and a 593-nm beam splitter, 624/40-nm emission filter, EM gain 25). Brightfield images were recorded for cell segmentation.

For comparing the flux-sensor output between co-existing high flux (dividing) and low-flux (non-dividing) cells, TM6 cells from log-phase (10 g/l glucose) cultures were loaded in the microfluidic device, and after one initial round of fluorescence imaging, cells were followed only with the brightfield channel for 20 h to determine budding activity. Cell segmentation to determine mean cell YFP and mCherry fluorescence intensities was performed with the ImageJ plugin BudJ (Ferrezuelo *et al*, 2012) using the brightfield images. Before computing the YFP to mCherry ratio for each cell, fluorescent intensity measurements were corrected for background fluorescence by subtracting the modal gray value of the whole image area in each fluorescent channel. Statistical analyses and plotting were performed in GraphPad.

In the experiments applying the sensor and unregulated control to the cell-cycle context, TM6 cells from log-phase cultures (20 g/l glucose) were loaded in the microfluidic device, where fresh 20 g/l glucose medium was provided at the flow rate of 4 μl/min with the help of a syringe pump. Microscopic imaging was performed in the YFP, mCherry, and brightfield channels every 6 min. Images in the fluorescent channels were background-corrected via rolling ball background subtraction plugin of ImageJ, and images in the brightfield channel were sharpened and contrast-enhanced in ImageJ. Cells were tracked throughout a microscopy movie and segmented by fitting an ellipse in the brightfield image at each time point via the semi-automated plugin BudJ (Ferrezuelo *et al*, 2012) used with ImageJ. In parallel, by visual inspection and with the help of a custom macro, we recorded for each segmented cell the time points of budding events (appearance of a dark-pixel cluster from which a daughter cell would later grow) and of cytokinesis events (one time point before the daughter cell would rapidly detach from the mother cell, sometimes accompanied by the appearance of a dark-pixel line between the mother and daughter cells). To analyze cellular fluorescence data, we uploaded the background-corrected microscopy movie into a NumPy multidimensional array via Python's module scikit-image and extracted the pixels corresponding to a cell of interest by overlapping the array with the segmentation ellipses provided by BudJ. To work with a continuous cell volume trajectory $V(t)${fl} without abrupt drops corresponding to cytokineses, we considered a cell cycle to be confined between two cytokineses (cyt), excluding the first but including the last one: $t \in (cyt_i, cyt_{i+1})$min. We calculated the volumes of the mother and daughter cells $V^m$ and $V^d$ separately, using the radii of the ellipse that ImageJ's plugin BudJ fitted to the mother and daughter cells in the brightfield image. We assumed that the mother and daughter cells are prolate spheroids;

therefore, $V^m$ and $V^d$ were calculated via $\frac{4}{3}\pi Rr^2$, where $R$ and $r$ are the major and minor radii, respectively. Given the microscope's resolution, it was infeasible to accurately segment daughter cells with BudJ for several time points after budding. In these time points, the daughter cell volume was reconstructed using linear interpolation between the zero volume at budding and the first volume calculation on the basis of BudJ-derived radii. Eventually, a cell-cycle trajectory of the cell volume was assembled as follows: $V(t) = V^m(t) + V^d(t), t \in (cyt_i, cyt_{i+1})$, with $V^d(t)$ equal to zero until budding. We visually inspected the cell-cycle trajectories of cell volume as well as YFP and mCherry fluorescence averaged across the mother-cell pixels, and removed those cell cycles from the analysis that suffered from bad cell segmentation and noisy signals. In some cell cycles, we also removed single data points corresponding to abnormally high or low cell volume explained by bad cell segmentation. We smoothed the cell volume, YFP, and mCherry trajectories to filter out local fluctuations caused by imperfect segmentation and to capture visible global behavior. To support smoothing at the beginning and end of a cell-cycle trace, we used the data in the adjacent 50 min of the preceding and following cell cycles. The smoothing was performed via the Gaussian process regression (*Python*'s *sklearn.gaussian_process*) using as a prior the radial basis function kernel with the length-scale range [30,48] {min} and the white kernel with the free noise level, and maximizing the log-marginal likelihood. The smoothed trajectories of cell volume, mother-cell YFP, and mCherry fluorescence, $V^{smooth}(t)$, $F^{smooth}_{YFP}(t)$, and $F^{smooth}_{mCherry}(t), t \in (cyt_i - 50, cyt_{i+1} + 50)$, were used to calculate the YFP and mCherry abundances $A_X(t) = V^{smooth}(t)F^{smooth}_X(t), X \in \{YFP, mCherry\}$. In turn, these abundances were employed to calculate the production rates in the first-order maturation kinetics model: $r_X(t) = \frac{t_{1/2}(X)}{ln2} \cdot \frac{d^2 A_X(t)}{dt^2} + \frac{dA_X(t)}{dt}$, $t \in [cyt_i, cyt_{i+1}]$, where $t_{1/2}(X)$ is the maturation half-time assumed to be 20 min for YFP and 50 min for mCherry (varying these parameters around these values did not markedly influence the results of the Fig 7F and G). Next, we detrended $r_X(t) \rightarrow \bar{r}_X(t)$ by subtracting the line connecting $r_X(cyt_i)$ and $r_X(cyt_{i+1})$, and normalized this trajectory so that its minimum becomes 0 and maximum 1: $\tilde{r}_X = \frac{\bar{r}_X(t) - \min(\bar{r}_X(t))}{\max(\bar{r}_X(t)) - \min(\bar{r}_X(t))}$. To determine the uncoupling between the YFP and mCherry production rates during the cell cycle, we calculated the following difference: $\tilde{r}_{YFP}(t) - \tilde{r}_{mCherry}(t)$. This analysis was implemented in *Python* with the help of the modules *pandas*, *numpy*, *scipy*, *sklearn*, *matplotlib*, *seaborn*, and *skimage* in the *Jupyter notebook* environment.

## Data availability

The datasets produced in this study are available in the following databases:

- Mass spectrometry raw data: ProteomeXchange Consortium PXD012964, http://proteomecentral.proteomexchange.org (via the PRIDE partner repository (Vizcaíno *et al*, 2013).
- Plasmids have been deposited at Addgene (plasmids #124582, #124583, #124584, and #124585).

Expanded View for this article is available online.

## Acknowledgements

The research of MH, GH, and FM has received funding from the European Commission (EC) under grant agreement no. 613745, PROMYS, the one of MH and JS under grant agreement no. 675585, Symbiosys, the one of MH and VT under grant agreement no. 642738, MetaRNA, and the one of MH, and AL under grant agreement no 289995, ISOLATE. Further, the authors would like to thank Marta Pałka and Ivan Frak for their support during early measurement campaigns and cloning, and Brenda Bley Folly and Alvaro Ortega for their advice on the implementation of the EMSA and thermal shift experiments.

## Author contributions

GH and MH conceived the idea of the study. FM, GH, and MH designed the study. FM, GH, JN, VT, SRV, and AL performed the experiments. HJW performed the protein modeling analysis. JH and AS performed the proteomic analysis. JS performed the computational metabolic flux analysis, and FM, GH, JN, VT, SRV, and AL analyzed experimental data. FM, GH, AL, VT, and MH wrote the manuscript. MH supervised the study.

## Conflict interest

The authors declare that they have no conflict of interest.

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
