## [Review Process File · Molecular Systems Biology]

Measuring glycolytic flux in single yeast cells with an orthogonal synthetic biosensor

Francisca Monteiro, Georg Hubmann, Vakil Takhaveev, Silke R. Vedelaar, Justin Norder, Johan Hekelaar, Joana Saldida, Athanasios Litsios, Hein J. Wijma, Alexander Schmidt, Matthias Heinemann.

Review timeline:

Submission date:	21 st June 2019
Editorial Decision:	31 st July 2019
Revision received:	25 th November 2019
Editorial Decision:	28 th November 2019
Revision received:	28 th November 2019
Accepted:	29 th November 2019

Editor: Maria Polychronidou

Transaction Report:

1st Editorial Decision

31st July 2019

Thank you again for submitting your work to Molecular Systems Biology. We have now heard back from the three referees who agreed to evaluate your study. As you will see below, the reviewers acknowledge that the presented biosensor seems potentially promising. They raise however a series of concerns, which we would ask you to address in a major revision.

I think that the reviewers' recommendations are rather clear and there is therefore no need to repeat the comments listed below. Most of the reviewers' concerns refer to the need to include further controls and analyses to better support the main conclusions. As reviewer #2 mentions, some level of follow up demonstrating the potential of the biosensor to reveal new biological insights that would not be possible to obtain using existing approaches would significantly enhance the impact of the study.

All other issues raised by the reviewers need to be satisfactorily addressed. As you may already know, our editorial policy allows in principle a single round of major revision so it is essential to provide responses to the reviewers' comments that are as complete as possible. Please feel free to contact me in case you would like to discuss in further detail any of the issues raised by the reviewers.

REFeree REPORTS

Reviewer #1:

Summary

This manuscript describes the development of a FBP biosensor that serves as a proxy of the glycolytic flux in yeast. Since FBP is a flux signaling metabolite (i.e. its concentration correlates to the glycolytic flux) the authors designed a transcription factor based system that responds to different FBP concentrations within yeast physiological range.

The authors modified the yeast CYC1 promoter to respond to *B. subtilis* CggR transcriptional factor and placed it upstream of the YFP fluorescent reporter. The CggR TF naturally binds FBP, in the absence of FBP it acts as a transcriptional repressor, while in the presence of FBP it can no longer bind the promoter and thus repress. The CggR heterologous TF was expressed at constant levels using a low expression mutant variant of the TEF1 promoter. To be able to detect the changes on the glycolytic flux in *S. cerevisiae* they engineer the CggR TF to bind FBP within the concentration range observed in *S. cerevisiae*. Finally, the authors evaluated the biosensor performance by FACS and microfluidics combined with time-lapse microscopy .

Along the work, the authors used several experimental and computational methods for the rational design of the biosensor and to test its performance. They demonstrate that the biosensor is condition independent and responsive to the physiological FBP concentration in single living yeast cells, allowing them to identify yeast subpopulations with different glycolytic fluxes.

General remarks

In most cases the authors have presented technically sound data and interpreted them appropriately. The manuscript is easy to read and evaluate. The methodological development that they present is timely relevant. As the authors mention in the introduction, the flux quantification combining ¹³C-based intracellular concentration measurements and computational modeling are widely spread in the metabolomics and fluxomics community. The development of biosensors has the advantage that they not only can report single cell fluxes, revealing population dynamics or the concentration of a particular metabolite, but also that this measurements can be rapidly perform *in vivo* as opposed to the ¹³C-based intracellular flux analysis. There are few examples in the literature that demonstrate that it is feasible to engineer novel biosensors combining bio-parts from different species. Therefore, this work represents an important example on how to properly incorporate a hybrid biosensor into a new host. It can be expected that the biosensor here presented will be use by other scientist to answer fundamental aspects of metabolism or to engineer yeast metabolism for industrial applications.

Major points

1) For the CggR promoter design, the authors performed a series of modifications to generate a CggR responsive promoter. The modifications that they performed aimed to reduce the accessibility to TATA boxes and eliminate the binding of CYC1 natural TF. Based in a computational prediction to minimize the nucleosome positioning over the promoter, further modifications were included in the promoter. The author demonstrate that the YFP levels driven by the synthetic promoter are condition independent and although the final outcome is satisfactory, the relevance of the modifications performed i.e. *cis* regulatory elements vs nucleosome positioning is uncertain. I suggest to experimentally demonstrate the impact of the modification comparing the YFP levels using the wt promoter, a promoter only including the *cis* regulatory modifications and the final promoter (*cis* element mutations plus minimized nucleosome positioning).

2) While the authors accounted for the autofluorescence in the FACS experiments they did not consider the possible fluorescence spectral overlap. Therefore, I would suggest them to compensate the FACS data. This reference could be useful <https://doi.org/10.1111/j.1749-6632.1993.tb38775.x>

4) The authors demonstrate that the sensor was able to distinguish subpopulations of isogenic cells with high or low glycolytic flux by following individual cells with microscopy. However, they do not report if they could distinguish these subpopulations by FACS . I suggest to re-analyze the FACS and report whether is possible or not.

Minor points

- 1) Along the manuscript, the author make a strong emphasis on the relevance of measuring the glycolytic flux at single cell level. I recommend that they also highlighted the relevance of doing this measurements in vivo.
- 2) The authors should describe better the CggR mechanism of repression. Reference: DOI: 10.1093/nar/gkq334
- 3) Based on the results presented in Figure 6, the authors conclude that the system with the CggR optimized version displayed a dynamic response that better covered the physiological concentration range of FBP than the system with the wt CggR. However, Figure 6 A and B suggest that while the system with the optimized CggR better resolves high FBP concentration, the system with the wt CggR sensor reports better low FBP concentration. I suggest to consider this when describing the results and the conclusion sections.

Reviewer #2:

The present manuscript develops a genetic reporter for FBP in yeast. The interest level in the topic and quality of the progress seem on track for MSB if 3 major concerns are adequately addressed:

1. Glycolytic flux is indirectly inferred, not measured. This is promising work from an excellent group, but false claims like the title only do harm to the field. One can have an informed debate about the concept of "flux sensing" metabolites, but however that debate ends, it does not justify mis-labeling concentrations as fluxes. It is essential to rectify the paper to properly acknowledge that it is about single-cell metabolite (not flux) sensing, with the benefit that this particular metabolite may in many (but not all) cases correlate with glycolytic flux.
2. Along these lines, it seems appropriate to show some counterexamples where FBP and Glycolytic flux diverge. The most obvious example would be aldolase KO. A physiological example would likely be oxidative stress, where GAPDH impairment probably leads to high FBP with modest flux? It is certainly disappointing to see correlations drawn with only 5 or 6 data points, some of them more or less pure negative controls (obviously there is no meaningful glycolytic flux with pyruvate as the substrate). The authors should also be aware that correlation analysis is only appropriate for normal distributed data and gives inflated R2 for cases where there are a few high points and then mainly near zero (i.e. Fig 2A). As drawn, Fig 2B seems like it would probably also give a significant correlation but R2 and p-value are omitted. This same issue carries over into Fig 4 (only 3 growth rates tested, experimental conditions unclear) and Fig 6 (where more than 4 points would be appreciated).
3. The back end of the paper is insufficient. It seems like the paper just ends 2/3 of the way through. Some development of biology beyond the presented very minimalistic use of the sensor, ideally one that indicates the validity of the sensor in mixed populations (or makes a novel biological claim that could be falsified from an orthogonal strategy like genetics or sorting) would seem appropriate to me for the paper to be at MSB level.

Lesser/presentation concerns:

4. It would be nice for the table to give some sense of what these mutations actually did. Right now it's work for reader to align between the table and Fig 5.
5. Fig 5D is not readily readable due to the color choice.
6. The shift in Kd for the R250A mutant is really small. So is the shift in DNA-binding. Why does this lead to material changes in fluorescent output curve? At least some discussion of this is merited.
7. The paper has 80+ references, to me quite biased towards self referencing, and still omitting any references to some major groups who have worked on flux/metabolite correlation in yeast.

Reviewer #3:

In this study, the authors orthogonally express the *B. subtilis* CggR gene, which encodes an FBP-binding transcription factor, in yeast to develop an intracellular sensor for glycolytic rate. This work

is relevant to the study of metabolic heterogeneity in cell populations and in that respect is quite interesting. The sensor design is rational and elegant, however what is really measured is FBP levels and this may or may not always correlate with glycolytic rate and they should adjust their claims accordingly.

1. The CggR sensor, at face value, is a sensor for intracellular FBP levels. While the authors do perform assays to validate that intracellular FBP levels do correlate with glycolytic rate in yeast, the correlations are not strong. Additionally, it could potentially be misleading to readers, since the authors often refer to their sensor as a sensor of "glycolytic flux" and not of FBP. In general, the text (and definitely the title of the paper) should be changed to reflect that the CggR sensor binds to FBP, and that interpretation of glycolytic flux rates are based on the assumption that FBP concentrations and glycolytic rates are directly correlated across many metabolic conditions.

2. As for any sensor of an intracellular metabolite, care must be taken to ensure that the sensor itself does not alter the metabolism of the cell. Given that the CggR sensor binds to FBP, can the authors better characterize the effects of CggR expression on intracellular FBP levels and on glycolytic rates?

1st Revision - authors' response

25th November 2019

Reviewer #1:

Summary

This manuscript describes the development of a FBP biosensor that serves as a proxy of the glycolytic flux in yeast. Since FBP is a flux signaling metabolite (i.e. its concentration correlates to the the glycolytic flux) the authors designed a transcription factor based system that responds to different FBP concentrations within yeast physiological range.

The authors modified the yeast CYC1 promotor to respond to *B. subtilis* CggR transcriptional factor and placed it upstream of the YFP fluorescent reporter. The CggR TF naturally binds FBP, in the absence of FBP it acts as a transcriptional repressor, while in the presence of FBP it can no longer bind the promotor and thus repress. The CggR heterologous TF was expressed at constant levels using a low expression mutant variant of the TEF1 promotor. To be able to detect the changes on the glycolytic flux in *S. cerevisiae* they engineer the CggR TF to bind FBP within the concentration range observed in *S. cerevisiae*. Finally, the authors evaluated the biosensor performance by FACS and microfluidics combined with time-lapse microscopy.

Along the work, the authors used several experimental and computational methods for the rational design of the biosensor and to test its performance. They demonstrate that the biosensor is condition independent and responsive to the physiological FBP concentration in single living yeast cells, allowing them to identify yeast subpopulations with different glycolytic fluxes.

General remarks

In most cases the authors have presented technically sound data and interpreted them appropriately. The manuscript is easy to read and evaluate. The methodological development that they present is timely relevant. As the authors mention in the introduction, the flux quantification combining ¹³C-based intracellular concentration measurements and computational modeling are widely spread in the metabolomics and fluxomics community. The development of biosensors has the advantage that they not only can report

single cell fluxes, revealing population dynamics or the concentration of a particular metabolite, but also that this measurements can be rapidly perform in vivo as opposed to the ^{13}C -based intracellular flux analysis. There are few examples in the literature that demonstrate that it is feasible to engineer novel biosensors combining bio-parts from different species. Therefore, this work represents an important example on how to properly incorporate a hybrid biosensor into a new host. It can be expected that the biosensor here presented will be use by other scientist to answer fundamental aspects of metabolism or to engineer yeast metabolism for industrial applications.

We would like to thank the reviewer for highlighting the relevance of our work.

Major points

1) For the CggR promoter design, the authors performed a series of modifications to generate a CggR responsive promoter. The modifications that they performed aimed to reduce the accessibility to TATA boxes and eliminate the binding of CYC1 natural TF. Based in a computational prediction to minimize the nucleosome positioning over the promotor, further modifications were included in the promotor. The author demonstrate that the YFP levels driven by the synthetic promotor are condition independent and although the final outcome is satisfactory, the relevance of the modifications performed i.e. cis regulatory elements vs nucleosome positioning is uncertain. I suggest to experimentally demonstrate the impact of the modification comparing the YFP levels using the wt promotor, a promotor only including the cis regulatory modifications and the final promotor (cis element mutations plus minimized nucleosome positioning).

To demonstrate the relevance of the introduced steps in the development of the synthetic promotor, as suggested by the reviewer, we now generated data that shows the relevance of the performed modifications. Specifically, we now include tests of the following three additional promotors (i) – (iii); (iv) was already included in the previous manuscript:

- i) The wildtype CYC1 core promotor before the introduction of the CggR cis-regulatory elements for the transcription factor CggR, i.e. the original sequence for the promotor upstream of the CYC1 gene
- ii) The CYC1 core promotor with the introduced cggRO elements (i.e. where at several points in the CYC1 sequence the CggRO binding sites were inserted)
- iii) The CYC1 core promotor with the introduced cggRO elements, after the initialization run of the optimization algorithm used for the nucleosome positioning (PMID 24862902).
- iv) The CYC1 core promotor with the introduced cggRO elements, after 38 optimization rounds toward reduced nucleosome affinity. This sequence differs in 37 positions from 3), and this is the promotor that we had already shown in the previous manuscript.

For this revised manuscript, we generated reporter constructs for the promotors (i) to (iii) and analysed the expression. Here, we found that expression of the first three variants was only marginally above the background fluorescence, while the variant with the minimized nucleosome affinity showed high expression levels. These data nicely underline the need for the optimization of the nucleosome affinity, which is indeed a crucial piece of information for others, aiming to develop similar synthetic promotors. We would like to thank the reviewer for having suggested this

experiment, which we have added to Figure 6, as subfigure A-C. A new paragraph was added to describe these results (2nd paragraph in section “Testing the glycolytic flux sensor”).

2) While the authors accounted for the autofluorescence in the FACS experiments they did not consider the possible fluorescence spectral overlap. Therefore, I would suggest them to compensate the FACS data. This reference could be useful <https://doi.org/10.1111/j.1749-6632.1993.tb38775.x>

Spectral overlap can indeed be an issue in quantitative flow cytometric studies. Here, we have chosen two fluorescent proteins (eCitrine and mCherry) that have hardly any spectral overlap when measured with our flow cytometer’s bandpass filters, as can be seen in the figure below showing the emission/excitation spectra of the two used fluorescent proteins and the range of the employed bandpass filters for the two emission channels.

Given this only very marginal overlap, we think that a spectral compensation would be a bit of an “overkill”. Yet, we have added the figure below as Appendix Figure S4, and we have added a statement on the topic to the revised version of the manuscript (first paragraph in the section “Testing the glycolytic flux sensor”).

3) The authors demonstrate that the sensor was able to distinguish subpopulations of isogenic cells with high or low glycolytic flux by following individual cells with microscopy. However, they do not report if they could distinguish these subpopulations by FACS. I suggest to re-analyze the FACS and report whether is possible or not.

The fraction of the quiescent subpopulation (i.e. the subpopulation with the low glycolytic flux) is $3.06\% \pm 1.93\%$ (mean \pm SD) of the total cells, as determined by microscopy. In the microscopy experiment, which we showed in the previous version of the manuscript, we could identify these cells on the basis of the fact that they phenotype did not divide for at least 40 hours despite the presence of glucose.

To identify a such small subpopulation, where there is only a 2-fold difference in the eCitrine/mCherry ratio between the two cell populations (Fig. 7D, as determined with microscopy), is challenging. Potentially, we might see these cells with flow cytometry (i.e. the cells in the B-gate in the right figure below; green fluorescence (FL1-A), the red fluorescence (FL3-A) of a population of TM6 cells. The B-gate contains 3% of the cells. However, we feel that this is too speculative and we would like to refrain from putting this in the manuscript.

Yet, encouraged by this reviewer's comment, we performed an alternative analysis, where we mixed wildtype and TM6 cells, both grown in glucose, in different proportions and performed flow cytometric analyses. Here, we found that with the FBP-difference as present between these two strains, we can safely determine subpopulations from 5% onward. We have added these data to the revised version of the manuscript (as new Figures 7A, B; 1st paragraph in section "Application of the sensor" is new). We would like to thank the reviewer for having triggered these analyses.

Minor points

1) Along the manuscript, the author make a strong emphasis on the relevance of measuring the glycolytic flux at single cell level. I recommend that they also highlighted the relevance of doing this measurements in vivo.

Throughout the manuscript we now emphasize that the glycolytic flux can now be measured in vivo, i.e. in living cells. In fact, the newly added experiment (i.e. resolving the glycolytic flux throughout the cell cycle) is a nice illustration of this.

2) The authors should describe better the CggR mechanism of repression. Reference: DOI: 10.1093/nar/gkq334

The mechanism of action of CggR is now described in more detail in the article, and we included the reference (cf. first paragraph in the section "Design of biosensor concept").

3) Based on the results presented in Figure 6, the authors conclude that the system with the CggR optimized version displayed a dynamic response that better covered the physiological concentration range of FBP than the system with the wt CggR. However, Figure 6 A and B suggest that while the system with the optimized CggR better resolves high FBP concentration, the system with the wt CggR sensor reports better low FBP concentration. I suggest to consider this when describing the results and the conclusion sections.

We agree. We now mention this point (cf. last paragraph of the section "Testing the glycolytic flux sensor"). (Note that Figure 6A and B are now Figures 6D and E).

Reviewer #2:

The present manuscript develops a genetic reporter for FBP in yeast. The interest level in the topic and quality of the progress seem on track for MSB if 3 major concerns are adequately addressed:

1. Glycolytic flux is indirectly inferred, not measured. This is promising work from an excellent group, but false claims like the title only do harm to the field. One can have an informed debate about the concept of "flux sensing"

metabolites, but however that debate ends, it does not justify mis-labeling concentrations as fluxes. It is essential to rectify the paper to properly acknowledge that it is about single-cell metabolite (not flux) sensing, with the benefit that this particular metabolite may in many (but not all) cases correlate with glycolytic flux.

We agree with the reviewer that the sensor cannot *directly* measure flux, as flux is only inferred from the levels of FBP. In this context, please note that also “¹³C flux measurement” does not directly measure flux, as it also only infers fluxes from measurements of state variables (i.e. measured ¹³C labelling patterns). Thus, strictly spoken, “flux measurement with ¹³C flux analysis” would also reflect incorrect claims.

We fear that if we would change the title to stating that we developed a sensor for FBP concentration we would significantly reduce the impact of the work. Reason: many researchers are interested in glycolysis and its activity/flux, and thus are attracted to look at the paper with its current title. In contrast, if we only state that we now can sense FBP concentrations, all those researchers, who are not aware of the correlation between flux and FBP levels, would not be attracted to read this paper. Thus, we are confronted with a ‘dilemma’.

Notably, in the abstract, we clearly mention that for the sensor we exploit the correlation between FBP and glycolytic flux. Thus, the reader is essentially left in the “certain” for only a few sentences. In the new abstract and the manuscript, we made it more clear that we measure a metabolite concentration, from which we can learn something about flux, because this metabolite concentration correlates with flux. We sincerely hope that the reviewer understands our reasoning and agrees with the textual changes. Alternatively, one could think about changing the title into “Measuring fructose-1,6-bisphosphate, and thus glycolytic flux, in single yeast cells with an orthogonal synthetic biosensor”.

2. Along these lines, it seems appropriate to show some counterexamples where FBP and Glycolytic flux diverge. The most obvious example would be aldolase KO. A physiological example would likely be oxidative stress, where GAPDH impairment probably leads to high FBP with modest flux? It is certainly disappointing to see correlations drawn with only 5 or 6 data points, some of them more or less pure negative controls (obviously there is no meaningful glycolytic flux with pyruvate as the substrate). The authors should also be aware that correlation analysis is only appropriate for normal distributed data and gives inflated R² for cases where there are a few high points and then mainly near zero (i.e. Fig 2A). As drawn, Fig 2B seems like it would probably also give a significant correlation but R² and p-value are omitted. This same issue carries over into Fig 4 (only 3 growth rates tested, experimental conditions unclear) and Fig 6 (where more than 4 points would be appreciated).

This comment contains two aspects:

First, the reviewer asks for a counter example, where FBP and glycolytic flux do not correlate. We are a bit uncertain of what efforts towards finding such counter examples would add to our work. In general, please note we did not mean to establish the correlation between FBP and glycolytic fluxes, which was presented in several cases before by us and others (e.g. PMID 21205161, PMID 30963997). The reason why we re-determined these data was to reaffirm the correlation for the

conditions, in which we here demonstrate the functionality of the sensor. Yet, it can indeed be that the correlation does not hold under all conditions/circumstance. Therefore, we have added a statement at the beginning of the discussion section pointing to this limitation.

To address the question on the significance, we performed additional analyses. Specifically, instead of considering the six data points that correspond to mean values, we utilized the variability of FBP-concentration and growth-rate measurements as well as the uncertainty in the model-based glycolytic-flux estimation. In specific, we considered the replicate measurements separately (i.e. not in the form of the mean as in the previous version) and the distributions of flux estimations (i.e. not a single values as in the previous version). We used multiple sampling with replacement from this dataset (bootstrapping) to calculate at each iteration a Pearson's correlation coefficient. This procedure allowed us to calculate confidence intervals for the correlation coefficients and p-values. We these analyses for both Figure 2A and 2B, and included the results in the revised version of the manuscript. The determined correlation coefficient and its confidence interval demonstrates that the FBP level and glycolytic flux are strongly linearly correlated. For Fig. 4, we now specify the experimental conditions in the caption. We are sorry for not having done this earlier.

3. The back end of the paper is insufficient. It seems like the paper just ends 2/3 of the way through. Some development of biology beyond the presented very minimalistic use of the sensor, ideally one that indicates the validity of the sensor in mixed populations (or makes a novel biological claim that could be falsified from an orthogonal strategy like genetics or sorting) would seem appropriate to me for the paper to be at MSB level.

For this revised version of the manuscript, we used the sensor for two additional experiments and analyses:

- First, along the lines of the reviewer's suggestion on the mixed populations, we mixed TM6 and WT, both grown on glucose, and investigated the subpopulation fraction that – given the glycolytic flux difference between these two strain –is minimally needed to resolve such subpopulations by flow cytometry. Here, we found a clear bimodal distribution at similar proportions and we found that a subpopulation needs to be around at least 5% to be identified by flow cytometry as a separate population. This experiment nicely complements the previous shown microscopy-based experiment. We have added the results from the new analysis as Fig. 7A, B. The first paragraph in the section “Application of the sensor” describes these results.
- Further, we performed additional experiments, demonstrating the applicability of the sensor also for slow dynamic cases. Particularly, we used the sensor to resolve the temporal behaviour of the FBP level/glycolytic flux during the cell cycle, an application for which, otherwise, synchronized cultures are needed. Specifically, we used microfluidics-assisted time-lapse microscopy to continuously monitor the cells expressing the sensor (and a control strain lacking CggR) and to track the YFP and mCherry signals as well as cell volume in single cells. Using these trajectories, we derived the production rates of YFP and mCherry during the cell cycle (cf. details in the M&M). As a proxy for momentary FBP and glycolytic flux, we used the uncoupling between the YFP and mCherry production rates. Here, we discovered that the FBP level and glycolytic flux are high in the G1 cell cycle phase, but drop after budding in the

middle of S-G2-M phase. We added these data as Fig. 7G-F and Appendix Figure S10. The results are described in the last two paragraphs of the section “Application of the sensor”.

We are grateful that the reviewer encouraged us to further push the sensor’s boundaries. We feel that it was absolutely worth to perform these experiments, particularly that we now could also show that we can use the sensor to resolve slow dynamic changes.

Lesser/presentation concerns:

4. It would be nice for the table to give some sense of what these mutations actually did. Right now it's work for reader to align between the table and Fig 5.

We have now included a table with a comparison between the expected/predicted behaviour with the actually observed one as Appendix Table S4.

5. Fig 5D is not readily readable due to the color choice.

The chosen colour scheme was indeed somewhat problematic for colour blind readers. We have now translated the red/green colours in -/+ signs to indicate undesired/desired features of the CggR mutants.

6. The shift in Kd for the R250A mutant is really small. So is the shift in DNA-binding. Why does this lead to material changes in fluorescent output curve? At least some discussion of this is merited.

Yes, the shift in the Kd for the mutant is small. Yet, as the Kd of the transcription factor is exactly in the physiological range of FBP concentrations, small Kd changes in this range will necessarily lead to different CggR occupancy with FBP and thus a different sensor output. We added a respective statement to the 2nd last paragraph in the section “Testing the glycolytic flux sensor”.

7. The paper has 80+ references, to me quite biased towards self referencing, and still omitting any references to some major groups who have worked on flux/metabolite correlation in yeast.

We have added more references to the introduction, which described data on flux/metabolite correlation in yeast. If we still have overlooked any further key papers, we would kindly ask the reviewer to provide us with the papers that we have missed.

Reviewer #3:

In this study, the authors orthogonally express the *B. subtilis* CggR gene, which encodes an FBP-binding transcription factor, in yeast to develop an intracellular sensor for glycolytic rate. This work is relevant to the study of metabolic heterogeneity in cell populations and in that respect is quite interesting. The sensor design is rational and elegant, however what is really measured is FBP levels and this may or may not always correlate with glycolytic rate and they should adjust their claims accordingly.

1. The CggR sensor, at face value, is a sensor for intracellular FBP levels. While the authors do perform assays to validate that intracellular FBP levels do correlate with glycolytic rate in yeast, the correlations are not strong. Additionally, it could potentially be misleading to readers, since the authors often refer to their sensor as a sensor of "glycolytic flux" and not of FBP. In

general, the text (and definitely the title of the paper) should be changed to reflect that the CggR sensor binds to FBP, and that interpretation of glycolytic flux rates are based on the assumption that FBP concentrations and glycolytic rates are directly correlated across many metabolic conditions.

We agree with the reviewer that the sensor cannot *directly* measure flux, as flux is only inferred from the levels of FBP. In this context, please note that also “¹³C flux measurement” does not directly measure flux, as it also only infers fluxes from measurements of state variables (i.e. measured ¹³C labelling patterns). Thus, strictly spoken, “flux measurement with ¹³C flux analysis” would also reflect incorrect claims.

We fear that if we would change the title to stating that we developed a sensor for FBP concentration we would significantly reduce the impact of the work. Reason: many researchers are interested in glycolysis and its activity/flux, and thus are attracted to look at the paper with its current title. In contrast, if we only state that we now can sense FBP concentrations, all those researchers, who are not aware of the correlation between flux and FBP levels, would not be attracted to read this paper. Thus, we are confronted with a ‘dilemma’.

Notably, in the abstract, we clearly mention that for the sensor we exploit the correlation between FBP and glycolytic flux. Thus, the reader is essentially left in the “certain” only for a few sentences. In the new abstract and the manuscript, we made it more clear that we measure a metabolite concentration, from which we can learn something about flux, because this metabolite concentration correlates with flux. We sincerely hope that the reviewer understands our reasoning and agrees with the textual changes. Alternatively, one could think about changing the title into “Measuring fructose-1,6-bisphosphate, and thus glycolytic flux, in single yeast cells with an orthogonal synthetic biosensor”.

2. As for any sensor of an intracellular metabolite, care must be taken to ensure that the sensor itself does not alter the metabolism of the cell. Given that the CggR sensor binds to FBP, can the authors better characterize the effects of CggR expression on intracellular FBP levels and on glycolytic rates?

We monitored the growth rate as a proxy for changed physiological behaviour due to the presence of the sensor itself. In all conditions, we saw that the presence of the sensor circuit (CggR and reporter plasmid/genes) did not alter the growth rate (Appendix Figure S5). Also, from a theoretical point of view, we do not expect any significant effect from titrating away FBP by CggR: Cellular CggR copies are in the order of a few 100s (as estimated from our proteome analysis), while FBP copies are in the order of 50-200 million copies per cell (as estimated from the data shown in Fig. 2A, from the average yeast cell volume of 40 fL, and the Avogadro number). Thus, a titrating effect, if it exists, would only be very marginal. We added sentence on this topic in the third paragraph in the section “Testing the glycolytic flux sensor”.

Thank you for sending us your revised study. We think that the additional analyses and the performed revisions have strengthened the study and we are pleased to inform you that it is now suitable for publication in *Molecular Systems Biology*.

Before we can formally accept the study for publication, we would ask you to address the following remaining editorial issues.

Corresponding Author Name: Matthias Heinemann

Manuscript Number: MSB-19-9071